# Modelling Microbial Communities with Graph Neural Networks

## Abstract

Understanding the interactions and interplay of microorganisms is a great challenge with many applications in medical and environmental settings. In this work, we model bacterial communities directly from their genomes using graph neural networks (GNNs). GNNs leverage the inductive bias induced by the set nature of bacteria, enforcing permutation equivariance and granting combinatorial generalization. We propose to learn the dynamics implicitly by directly predicting community relative abundance profiles at steady state, thus escaping the need for growth curves. On two real-world datasets, we show for the first time generalization to unseen bacteria and different community structures. To investigate the prediction results more deeply, we create a simulation for flexible data generation and analyze effects of bacteria interaction strength, community size, and training data amount.

## 1 Introduction

Microorganisms are ubiquitous and essential: in our gut, they digest our food and influence our behavior (Cani et al., 2019); in industrial plants, they treat our wastewater (Mathew et al., 2022); their biomining ability outside of Earth was even tested on the International Space Station (Cockell et al., 2020). Accordingly, understanding their functioning and optimizing their use are crucial challenges.

Microbial communities are driven by interactions that dictate the assembly of communities and consequently microbial output. To comprehend the functioning of a community, it is necessary to characterize these interactions. Ideally, one would acquire time-series data for every combination of bacteria to obtain a complete understanding of their dynamics. However, in reality, this is not possible because the number of experiments grows exponentially with the number of bacteria. Accordingly, several challenges are faced when modeling bacterial interactions: (i) available data generally depict a single time-point of a community; (ii) models of interactions should generalize to new bacteria and communities to limit the need for additional experiments; (iii) models should be interpretable and provide insights on the system.

The most common approach to model interactions in bacterial communities is to use generalized Lotka-Volterra models (Gonze et al., 2018; van den Berg et al., 2022; Picot et al., 2023) (gLV, see Sec. 2.1). However, these deterministic models fit parameters on time-series data for each bacterium in the system: therefore, they cannot generalize to new bacteria and are limited by experimental data. Furthermore, as they only model pairwise interactions, they may fail to recover higher-order/complex interactions (Chang et al., 2023; Gonze et al., 2018; Picot et al., 2023). However, it should be noted that there is a debate in the field about whether bacterial communities are shaped by simple (Friedman et al., 2017; Goldford et al., 2018) or complex (Bairey et al., 2016; Chang et al., 2023) assembly rules. To address the potential complexity of microbial systems, neural networks are emerging as alternatives to gLV models, as they can capture complex interactions (Baranwal et al., 2022; Michel-Mata et al., 2022). For instance, Baranwal et al. (2022) fit recurrent neural networks to microbial communities of up to 26 bacteria to predict their assembly and ultimately a function of interest, namely butyrate production. Although their results are encouraging, their models are fitted on growth trajectories and rely on time-series, impeding their generalization to new bacteria and communities.

In this work, we model bacterial communities directly from bacterial genomes using graph neural networks (GNNs). Our contribution can be described as follows.

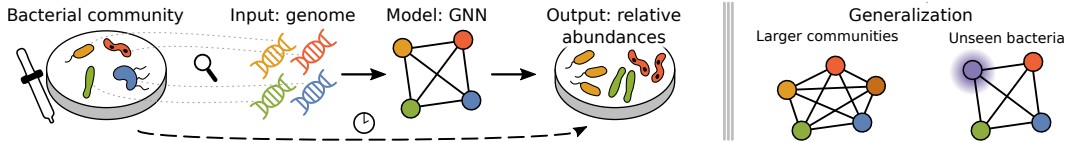

Figure 1: We propose to leverage Graph Neural Networks to implicitly learn bacterial communities' dynamics from bacterial genomes. This method allows accurate predictions of steady-state community profiles and generalization to larger communities and unseen bacteria.

1. We propose using GNNs as a powerful class of function approximators to model microbial communities, such that each node in the graph represents a bacterial species and the GNN performs regression on nodes. Through the graph structure, GNNs isolate information and share parameters across nodes, thus granting permutation equivariance and generalization to unseen bacteria, and enabling the prediction of compositions of microbial communities.

2. We explore learning community dynamics directly from genomes: since nucleic acids are the universal information carrier of living organisms, this can, in principle, allow generalizing to any unseen microorganisms.

3. We propose learning dynamics implicitly by directly predicting community relative abundance profiles at steady state, thus escaping the need for growth curves.

4. We propose a simulation framework to facilitate exploratory benchmarks for models of microbial communities using genome features.

In practice, we evaluate the ability of conventional architectures (i.e. MLPs) and GNNs to model bacterial communities on two publicly available datasets (Friedman et al., 2017; Baranwal et al., 2022), and further explore hypotheses in simulations. Our results show that GNNs can accurately predict the relative abundances of bacteria in communities from their genomes for communities of various compositions and sizes. Furthermore, GNNs can generalize to marginally bigger communities and new bacteria not seen during training.

## 2 METHODS

### 2.1 TERMINOLOGY AND PROBLEM DEFINITION

**Bacterial communities**   A *bacterium*, plural *bacteria*, is a unicellular microorganism. Bacteria are classified via a taxonomy based on the DNA, the finer-grained groupings being the *genus*, *species*, and *strain*. The bacteria in one strain are clones with almost identical DNA. In this work, we will use the species designation to refer to different bacteria. A bacterial *community* is formed by two or more species of bacteria that grow in the same environment. A community can be described by a set $S$ of bacterial species. At any time $t$, each bacterial species $s_i \in S$ is present in the environment in *abundance* $n_i(t)$. We define $y_i(t) := n_i(t)/\sum_{j \in [1,|S|]} n_j(t)$ as the *relative abundance* of bacterium $s_i$ at time $t$. Over time, these metrics vary according to the properties of each species (e.g. growth rate), as well as complex inter-species interactions. Extrinsic factors may affect the amount of bacteria in the environment, for instance, the amount of resources, but we will ignore them for simplicity as in previous work (Bashan et al., 2016). This is especially justified in the case of experimental data from controlled environments (van den Berg et al., 2022).

**Generalized Lotka-Volterra model**   Our method learns to model community dynamics implicitly through a neural network and thus makes minimal modeling assumptions. Nevertheless, to give an intuition of how bacterial communities change over time, we now describe a simplified predictive model.

The generalized Lotka-Volterra model (Lotka, 1920; Volterra, 1926) describes the change in abundances in the environment (van den Berg et al., 2022; Gonze et al., 2018) according to

$$\frac{\mathrm{d}n_i}{\mathrm{d}t} = n_i(t) \cdot \mu_i \cdot \left(1 - \frac{1}{K_i} \sum_{j=1}^{|S|} a_{i,j} n_j(t)\right), \tag{1}$$

with $S$ the set of bacterial species in the environment. For a given species $s_i \in S$, $\mu_i$ is the growth rate and $K_i$ represents the carrying capacity, which limits the amount of bacteria that can exist in the environment. Finally, $a_{i,j}$ is an interaction factor describing the effect of species $s_i$ on species $s_j$, and $a_{i,i} = 1 \ \forall i \in [1, |S|]$.

**Genomes** Bacterial genomes consist of DNA sequences organized into genes, coding for all information related to bacterial functioning, e.g. metabolism, growth. Thus, genomes can be represented by the presence/absence of genes or groups of genes. An example of gene grouping is their mapping to the KEGG Orthology database to group them by molecular function (Moriya et al., 2007). For instance, the genome of *Anaerostipes caccae* carries the gene coding for the enzyme EC 1.3.8.1, which is a butyryl-CoA dehydrogenase belonging to the KEGG group KO K00248. Through the KEGG Orthology database mapping, genes coding for proteins with similar functions across species have the same annotation, and bacteria with similar molecular abilities have more similar representations.

In the context of this work, we represent genomes using feature vectors. Such vectors should have the same dimensionality and semantics across all bacteria. To represent all bacteria in a unified way, we consider all genes that occur in any genome in the pool of bacteria and record their presence/absence in the feature vector. Given an ordered set of $M$ genes $(g^k)_{k=0}^M$, we represent the genome of species $s_i \in S$ as a binary indicator vector $\mathbf{x}_i = (x_i^k)_{k=0}^M$ such that $x_i^k$ is one if gene $g^k$ is present in the genome of $s_i$, and zero otherwise. Hence, each bacterium, or node, has for attributes a binary vector representing the genome. For real data, the representation is taken from geanome annotations and for simulations the representation is abstracted to contain information on bacterial growth (see section 2.4).

**Task** Our aim is to predict the composition of bacterial communities $C \subseteq S$ at steady state from the genomes of the mixed bacteria. More specifically, we cast this task as a supervised learning problem. Assuming an equilibrium is reached at time-step $T$, our learning target is the observed relative abundance of each bacterial species $s_i \in C$ at equilibrium: $\mathbf{y}(T) = (y_1(T), \ldots, y_{|C|}(T))$. Our inputs are the feature vector representation of genomes of bacteria present in the mixture $\mathbf{x}_i \ \forall i \in [1, |C|]$. To compare architectures with fixed length input, namely MLPS, we add null feature vectors $\mathbf{x}_i = (0)_{k=0}^M$ for the bacteria absent from the mix.

## 2.2 MODELS

Our method learns an *implicit* model of the dynamics of a bacterial community. Instead of estimating the parameters of a differential equation, which can then be solved to retrieve an equilibrium, we apply a flexible class of function approximators and directly regress the solution at equilibrium. MLPs constitute a simple baseline, as they can in principle approximate arbitrary functions (Cybenko, 1989). As most commonly used neural network architectures, MLPs assume that the position of each input carries a semantic value. The prediction of bacterial community dynamics, however, has an interesting property, namely permutation equivariance. This is due to the fact that a community is a *set* of species, and the predictions of the model should not be affected by the order in which the species are presented. For this reason, we propose to leverage Graph Neural Networks (GNNs) (Scarselli et al., 2009; Gilmer et al., 2017; Kipf & Welling, 2017; Battaglia et al., 2018) to exploit this particular inductive bias.

GNNs can be formalized as Message Passing Neural Networks (MPNNs) (Gilmer et al., 2017). A graph is described by the tuple $G = (V, E)$, where $V$ denotes the set of vertices and $E$ the edges. The neighborhood of a vertex, i.e. node, $v \in V$ is described by $\mathcal{N}(v) = \{u | \{u, v\} \in E\}$. The attribute of each node is given by $\mathbf{x}_i$ for $i \in [1, |V|]$. In general, the attribute $\mathbf{x}_i$ of each node in the graph is updated as follows in each message passing step:

$$\mathbf{e}_{(i,j)} = g_e(\mathbf{x}_i, \mathbf{x}_j) \tag{2}$$

$$\mathbf{x}_i' = g_v(\mathbf{x}_i, \operatorname{aggr}_{j \in \mathcal{N}(i)}(e_{(i,j)})). \tag{3}$$

where $g_e$ and $g_v$ are arbitrary functions used for the edge and node computations respectively. The permutation-equivariant aggregation function is given by aggr. Depending on the choice of the node and edge update rules, we can recover different GNN architectures. In this work, we investigate two architectures: a spatial-convolutional GNN using the GraphSAGE implementation (Hamilton et al., 2017), and a slight variation of the message passing GNN architecture in Kipf et al. (2020), which we will refer to as MPGNN. The GATv2 (Veličković et al., 2018; Brody et al., 2022) and GCNII (Kipf & Welling, 2017; Chen et al., 2020) architectures were also tested but underperform the above models; see results in the Appendix A and Table S3. Given the lack of prior knowledge about the underlying graph topology, we use fully connected graphs such that each node is updated based on all other nodes within one message-passing step. The information propagation over $k$-hops can capture $k$-order relations between entities: the first message passing is limited to the neighboring

node attributes (pairwise interactions) and the next ones propagate the interactions of neighbors (bacterium $n_i$ receives information from $n_j$ and how it has been affected by others).

For `GraphSAGE`, the edge computation $\mathbf{e}_{(i,j)}$ returns the attributes of neighboring nodes $j \in \mathcal{N}(i)$, i.e. $g_e(\mathbf{x}_i, \mathbf{x}_j) = \mathbf{x}_j$. The node update function $g_v$ is given by: $\mathbf{x}'_i = W_1 \mathbf{x}_i + W_2 \cdot \text{mean}_{j \in \mathcal{N}(i)} \mathbf{x}_j$, where $W_1$ and $W_2$ are learnable parameters. The mean is used as the aggregation function. By using $k$ graph convolutional layers after one another, we can achieve $k$-hop information propagation. Finally, we have an additional linear layer at the end with sigmoid activation for the node attribute readouts.

In the `MPGNN`, we update the node attributes as $\mathbf{x}'_i = g_v(\mathbf{x}_i, \text{mean}_{j \in \mathcal{N}(i)}(g_e(\mathbf{x}_i, \mathbf{x}_j)))$. Here, $g_v$ and $g_e$ are MLPs with $l$ linear layers, each followed by a non-linearity, e.g. ReLU activation. Layer normalization is applied in the final layer. For the mapping from the node attributes to the outputs, we also have a linear layer with sigmoid activation. For `MPGNN`, $k$ message-passing steps are equivalent to the $k$-hop information propagation we get by stacking $k$ `GraphSAGE` layers. We treat $k$ as a hyperparameter for both `MPGNN` and `GraphSAGE`. For `MPGNN`, the number and size of the hidden layers of $g_e$ and $g_v$ are both tuned as hyperparameters, more details are given in Table S2.

Models were trained with the Adam optimizer (Kingma & Ba, 2015) to minimize the Mean Squared Error (MSE). We use the coefficient of determination $R^2$ to assess model performance on test set, with $R^2 = 1 - \frac{\sum_{i \in N}(x_i - \widehat{x}_i)}{\sum_{i \in N}(x_i - \widehat{x})}$. To allow calculating $R^2$ across communities, we center values with each community such that $\widehat{x} = 0$. We compute $R^2$ on 100 bootstraps of communities and report its average and 95% confidence interval; $R^2 = 1$ correspond to a perfect model, $R^2 \leq 0$ means the model is worse than random (i.e. predicting the mean). Implementation details, data splits, and reported metrics are detailed in Appendix A.

## 2.3 Publicly available real data

We use two publicly available datasets independently recorded by separate laboratories; we describe them here and provide more details in Appendix A.2. Networks were trained independently on each.

**Friedman2017** Experimental data from Friedman et al. (2017) consists of the relative abundances of 2, 3, 7, and 8-bacteria communities (Fig. S3, Fig. S4, and Fig. S5). The dataset contains 93 samples with 2 to 15 replicates each. Raw data was kindly provided by Friedman et al. (2017) and is now available on our project webpage `https://sites.google.com/view/microbegnn`.

**BaranwalClark2022** The dataset is published by Baranwal et al. (2022), with certain samples originally produced by Clark et al. (2021). The dataset is composed of relative abundances of 459 samples of 2 to 26-bacteria communities, each replicated 1 to 9 times. When testing generalization to excluded bacteria (see Sec. 3.3), we do not attempt to generalize to (i) *Holdemanella biformis* (HB) as the samples containing this bacterium are only present in two community sizes (2 and 26), resulting in a small test set, and (ii) *Coprococcus comes* (CC), *Eubacterium rectale* (ER), *Roseburia intestinalis* (RI), and *Faecalibacterium praustnitzii* (FP) due to their over-representation in samples, and so the resulting small training sets.

Genomes of bacterial species were downloaded from NCBI (Sayers et al., 2022) or the ATCC Genome Portal (Yarmosh et al., 2022), annotated with the NCBI prokaryotic genome annotation pipeline Tatusova et al. (2016), and genes were mapped to the KEGG database to obtain functional groups (Moriya et al., 2007). When a specific strain's genome was unavailable, the genome of the closest type strain was used instead. Details on strain genomes are provided in Supplementary Table S4. We used the presence/absence of each KO group as input for fitting models; KO annotations present in all genomes in a dataset were excluded.

## 2.4 Modeling bacterial communities in simulation

We design a simulator for the growth of bacterial communities based on the generalized Lotka-Volterra model (see Sec. 2.1), to control data parameters and specifically assess the application of GNNs to bacterial communities. This simulator, as illustrated in Fig. 2, is not meant to produce a faithful representation of real communities, but rather to provide a generative procedure that captures certain challenges in the data, e.g. large dimensionality, while controlling other characteristics, e.g. sample size.

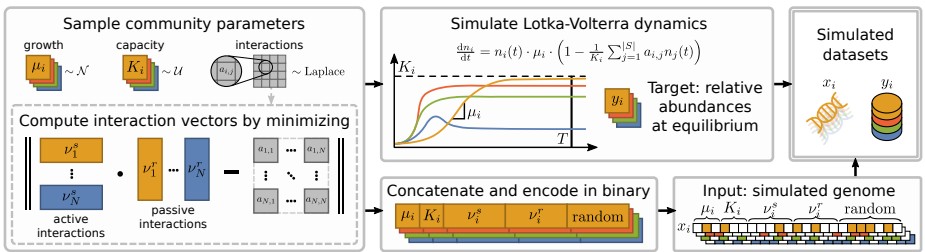

Figure 2: **Simulation of bacterial community growth dynamics and genomes.** For each bacterium $i$ in the community, we randomly draw its growth parameters: the growth rate $\mu_i$, the carrying capacity $K_i$, and its interaction factors $a_{i,\cdot}$. From these parameters, we calculate the abundances of bacteria at equilibrium according to the generalized Lotka-Volterra equations. We use the relative abundances of bacteria at equilibrium as a learning target. In parallel, we simulate genomes by creating binary vectors that encode for the growth parameters. For that, we first approximate two vectors per bacterium of $n_{\dim}$ dimensions: one to determine the effect of bacterium $i$ on others, $\nu_i^s$, and the other to determine the effect of other bacteria on $i$, $\nu_i^r$. For bacteria $i$ and $j$, we assume that the effect of $i$ on $j$ is expressed as $a_{i,j} = \nu_i^s \cdot \nu_j^r$. The approximation is performed by minimizing the distance between the real matrix of interactions and the one generated by the approximated vectors. We generate the binary encoding of the parameters $\mu_i$, $K_i$, and the values in $\nu_i^r$ and $\nu_i^s$. We also randomly draw a $[0-1]$ vector to add noise to genomes. We use these "genomes" as input features to predict the relative abundances of bacteria at equilibrium.

**Bacterial growth** The growth of each bacterium in the community was simulated using the generalized Lotka-Volterra equation (Eq. 1), with: $\ln(\mu_i) \sim \mathcal{N}(1, 0.5^2)$ clipped to $[0.5, 2]$, $K_i \sim \mathcal{U}(5, 15)$, and $a_{i,j} \sim \text{Laplace}(0, 0.2^2)$ clipped to $[-2, 2]$, $\forall i, j \in [1, |S|]$. The target relative abundance was calculated by simulating community growth until equilibrium: $n_i(0) = 10 \; \forall i \in [1, |S|]$ and equilibrium was reached when $\mathrm{d}n_i/\mathrm{d}t \leq 10^{-3} \; \forall i \in [1, |S|]$ (Fig. S1). Theoretically, this is similar to solving the roots of Eq. 1 which implies that steady-states depend on the parameters $K_i$ and $a_{i,j} \; \forall i, j \in [1, |S|]$. Given our set of parameters, all simulated communities were stable.

**Bacterial genomes** Bacterial genomes are generated to encode the simulated growth parameters such that there exists an approximately bijective mapping from genomes to parameters. We achieve this by rescaling parameters to $[0, 1]$, discretizing them, and performing a simple binary encoding to $n_g$ bits as $g_{\text{bin}} = \text{bin}\left(\text{round}\left((g - g_{\min})/(g_{\max} - g_{\min}) \cdot (2^{n_g} - 1)\right)\right)$. Although the encoding is not representative of any biological process, the mapping can be computed efficiently, provides a compact representation, and can be inverted up to discretization. This method is applied directly to the parameters $\mu$ and $K$, resulting in two binary vectors of size $n_g$.

Encoding the interaction factors $a_{i,j}$ into the genomes of each bacteria requires an additional step. Given a bacterial community $S$, two intermediate $n_{\dim}$-dimensional vectors for each bacterium $s_i \in S$ are needed: one determining its effect on interaction partners, $\nu_i^s \in \mathbb{R}^n$, and the other determining how it is affected by others, $\nu_i^r \in \mathbb{R}^n$. These vectors should contain sufficient information, such that the influence of bacterium $s_i$ on $s_j$ (encoded in $a_{i,j}$) can be retrieved from $\nu_i^s$ and $\nu_j^r$. For each pair of bacteria $(s_i, s_j) \in S^2$, we simply reconstruct interactions through inner products: $\hat{a}_{i,j} = \nu_i^s \cdot \nu_j^r$. We treat intermediate vectors as learnable parameters, and optimize them through gradient descent by minimizing the distance of the reconstructed interaction matrix from its ground truth: $J = \sum_{i \in [1, |S|]} \sum_{j \in [1, |S|]} (\hat{a}_{i,j} - a_{i,j})^2$. The $n_{\dim}$ vector coordinates for both vectors are finally encoded in the genomes as described above for $\mu$ and $K$.

Here, we use $n_g = 7$ for all parameters, $n_{\dim} = 20$ for the 25 simulated bacteria, and add 5 % of random genes. Empirically, we verify that $\mu$, $K$, and $\nu^s, \nu^r$ can be accurately recovered from simulated genomes.

## 3 EXPERIMENTS AND RESULTS

The general goal of this work is to train and evaluate neural models for the dynamics of bacterial communities, directly from their genomes. On real data (FRIEDMAN2017 and BARANWALCLARK2022),

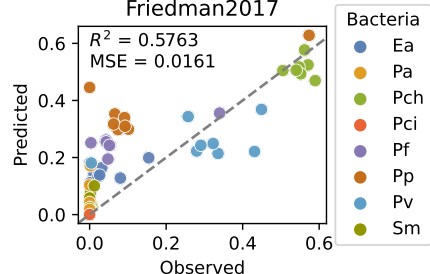

Figure 3: **Accuracy of MLP and GNN models on predictions of in-distribution and bigger bacterial communities.** Average $R^2$ values for predictions on in-distribution test sets with 95% confidence interval ($R^2 = 1$: perfect model, $R^2 \leq 0$: random model). The ensemble predictions across 5 model seeds were used and 5-folds CV was additionally used for **A**. **B**: Models were trained on communities of constrained sizes and tested on communities of distinct sizes.

we first investigate whether in-distribution predictions of unseen bacterial communities with known bacteria are possible. Then, we evaluate the generalization of learned models to (i) larger communities and (ii) unseen bacteria with respect to those used for training. Finally, due to the scarcity of real data, we leverage our proposed simulator to produce a dense and controllable distribution over communities: by retraining models on simulated data, we are able to validate whether trends emerging in real data can be explained in a simplified setting.

### 3.1 CAN WE MODEL REAL COMMUNITIES? — YES

We first set out to evaluate the general feasibility of predicting bacterial community profiles from bacterial genomes using GNNs (Fig. 3 A-B). Due to the set nature of communities, their dynamics are inherently permutation equivariant. This known property of the target function might however not be captured by universal function approximators such as MLPs. To confirm this, we train both GNNs and MLPs on the FRIEDMAN2017 dataset. When shuffling the order of bacteria within the train and test communities, the accuracy of MLPs drops significantly, clearly showing that the dynamics learned by MLPs are not equivariant to permutations (Fig. 3 A), and thus fundamentally incorrect. Both MPGNN and GraphSAGE provide accurate predictions. After some parameter tuning (see Supplementary Table S2), our best model predicts unseen bacterial mixes with a goodness of fit $R^2 = 0.8088$ and $R^2 = 0.7656$, for FRIEDMAN2017 and BARANWALCLARK2022 respectively (Fig. 3 A).

### 3.2 CAN WE GENERALIZE TO LARGER COMMUNITIES? — MARGINALLY.

We assess the ability of the models to generalize to communities of larger or smaller sizes. The motivation in the former case is to transfer knowledge from lab experiments on smaller communities to larger ones observed in the wild. In the latter case, the motivation is to evaluate whether one can learn a model from a large dataset of observed samples, and infer a model of bacterial interactions from it to monitor bacteria in the lab.

We train GNNs on communities with 2- and 3-bacteria and predict those with 7- and 8-bacteria from the FRIEDMAN2017 set. For the BARANWALCLARK2022 dataset, we train either on communities with 2- to 15- or 2- to 22-bacteria and predict the 23- to 26-bacteria communities (Fig. 3 B). The best ensemble models for each dataset have an accuracy of $R^2 = 0.5525$, $R^2 = 0.2606$, and $R^2 = 0.4486$, respectively (Fig. 4 and Fig. 5). For

Figure 4: **Prediction of larger communities in FRIEDMAN2017.** Each point represents a bacterium in a community of size $\geq 7$ bacteria; x-axis: average of the observed rel. abundance; y-axis: model predictions. Models trained on communities of size $\leq 3$.

BARANWALCLARK2022, including communities of sizes closer to test sizes greatly improves accuracy, suggesting that interactions may be different in larger communities, hence limiting the models' ability to generalize; we explore this hypothesis on simulated data in Sec. 2.4. This may explain why the models wrongly predict the growth of *Pseudomonas citronellolis* (Pci) and *Serratia marcescens* (Sm) in the FRIEDMAN2017 dataset (Fig. 4). Although in the observed communities, these bacteria do not survive, the models predict a significant abundance.

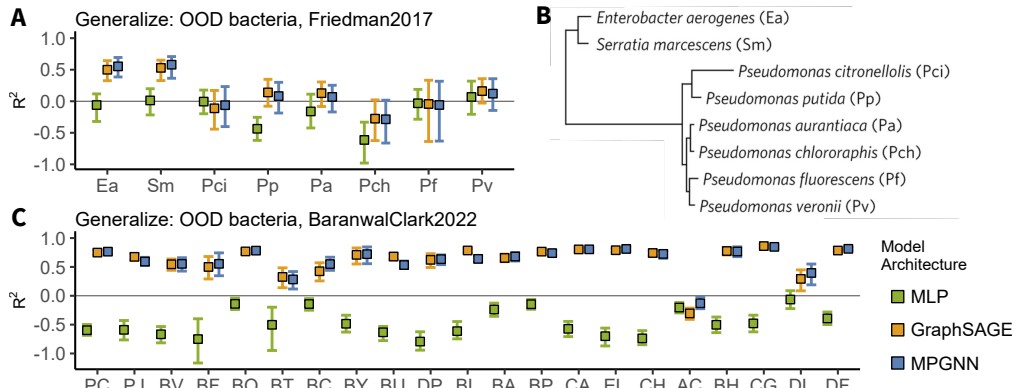

Figure 6: **Accuracy of GNN models on predictions of bacterial communities containing unseen bacteria.** For each bacterium, communities containing the bacterium are used for testing, and not for training. Average $R^2$ values on the test set are shown with 95% confidence interval (calculated on 100 sample bootstraps). **A-B**: Results and phylogenetic tree for FRIEDMAN2017. **C**: Results for BARANWALCLARK2022.

For the BARANWALCLARK2022 data, predictions on *Anaerostipes caccae* (AC) are the less accurate: the relative abundance of the bacterium is largely overestimated with an MSE = 0.0185 compared to MSE = 0.0006 for the other bacteria (Fig. 5). This difficulty to generalize to AC is consistent across our results (see Sec. 3.3). Training on larger communities to predict smaller ones does not achieve good results with all $R^2$ lower than zero, indicating worse accuracy than predicting the average (Fig. 3 B). Empirically, our results suggest that generalization to smaller communities poses different challenges with respect to generalization to larger communities.

### 3.3 CAN WE GENERALIZE TO UNKNOWN BACTERIA? — SORT OF.

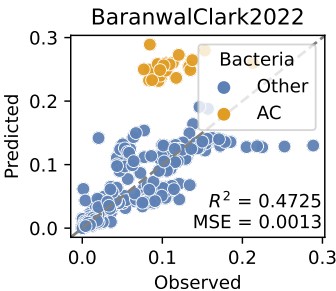

Figure 5: **Prediction of larger communities in BARANWAL-CLARK2022.** Models trained on communities of size $\leq 22$ bacteria, predictions for $\geq 23$.

Generalization to unseen bacteria is a challenging task that to our knowledge has not yet been performed for community growth dynamics. If successful, this suggests that models are able to extract relevant information from genomes that likely relate to biological processes causing the observed relative abundance of a bacterium in a community. This could open new possibilities, such as anticipating the effect of new pathogens on microbiomes or creating communities in an informed way by forecasting which bacteria is most likely to serve a desired purpose.

In practice, for every bacterium $s_i \in S$ we filter the training set to remove all communities that contain $s_i$, and use all communities that contain $s_i$ for testing. As no parameter tuning was performed, we do not use a validation set; results are shown for the test set directly.

The results vary depending on which species was left out as an unseen bacterium (Fig. 6). For instance, reasonable accuracies were obtained on the FRIEDMAN2017 dataset for predicting unseen bacteria *Enterobacter aerogenes* (Ea) and Sm (Fig. 6 A and Fig. S7 C; $R^2 = 0.5528$ and $R^2 = 0.5796$, respectively). Interestingly, these two bacteria were the most distant to the rest, being the only non-*Pseudomonas* (Fig. 6 B). A hypothesis is that they do not interact much with the *Pseudomonas*, or that they both interact in a similar manner. In line with this hypothesis, for *Pseudomonas*, growing with either Sm or Ea led to resembling communities, making it possible for the knowledge gained from the genome of one non-*Pseudomonas* to be accurately transferred to the other. This hypothesis is supported by the comparable relative abundances of *Pseudomonas* in 2- and 3-bacteria communities with Sm or Ea (Fig. S7 A). Predictions of communities with *Pseudomonas chlororaphis* (Pch) achieve the lowest accuracy, in fact lower than predicting the mean relative abundance for both types of models (Fig. 6 A, $R^2 < 0$). The genome of this species is not available on public databases, so the genome of the closest species had to be used instead. Hence, an uncontrolled error was introduced in the data. Furthermore, the substitute genome belongs to the

Figure 7: **Evaluation of prediction accuracy according to community features on simulated data.** Bacterial communities were simulated following the procedure described in Sec. 2.4 with varying parameters. MLP* receives each bacterium at a fixed input location. Bacteria not present have 0-vectors. MLP is a version with shuffled input at training time. **A**: The edge density, i.e. the probability of an edge, is varied between 0.1 and 0.6. **B**: The edge density is set to 0.2 except for 2 bacteria with an edge density of 0.8 to mimic keystone bacteria. Each keystone and five random bacteria are excluded from training as in Sec. 3.3, results are shown for the test set including communities with these bacteria. **C**: The training set is limited to communities of sizes below 10, 15, 25 bacteria; the test set contains communities of sizes 16 to 25. The number of training communities is reported below (size).

same species as *Pseudomonas aurantiaca* (Pa), which has a different phenotype than Pch in cultures, leading to different relative abundances in communities (Fig. S7 B). Nonetheless, models generalize better to Pa (Fig. 6 A, $R^2 = 0.1279$ for GraphSAGE). Hence, we can hypothesize that the models learn well from other *Pseudomonas* genomes, but cannot generalize well to Pch due to its substitute genome.

The results obtained on BARANWALCLARK2022 are superior to those on FRIEDMAN2017 data (Fig. 6 C). This could be attributed to the larger dataset size, which includes more bacteria and community sizes, thus providing a better resolution of the feature space (a wider range of genomes to learn from) and output space (more examples of co-cultures due to the increased number of communities). Nevertheless, we report significantly lower accuracy when generalizing to communities including AC. This bacterium is not particularly phylogenetically distant from others (Fig. S6), but is the only one that can produce butyrate from lactate and is a driver of butyrate production (Clark et al., 2021). Empirically, it inhibits the growth of CC, CH, BO, BT, BU, BC, and BY in communities of 11- to 13-bacteria while promoting the growth of CA and DL (Fig. S8 A; see the abbreviations in Supplementary Table S4). However, these effects are less clear in communities of 23- to 25-bacteria (Fig. S8 B). The other bacterium to which models can transfer less accurately is *Bacteroides thetaiotaomicron* (BT; Fig. 6 C). This bacterium is considered a keystone of the human gut microbiota, meaning that it drives community assembly (Banerjee et al., 2018). Consequently, communities including such a bacterium may be harder to predict due to the changes in interactions compared to communities without the bacterium, which explains the lower accuracy of the GNNs when generalizing to communities with BT. Actinobacteria, the phylum to which AC belongs, are also considered a keystone of the human microbiota (Banerjee et al., 2018). Although AC itself has not been reported to be a keystone, our results, together with the observation of butyrate production from Clark et al. (2021), suggests that it may be one. We explored this hypothesis on simulated data in Sec. 2.4.

Our results suggest that GNNs can generalize predictions of bacterial relative abundances to communities including unseen bacteria. In practice, the performance of models may still be limited due to noise in inputs (genomes) and output resolution (similar genomes but different phenotypes).

### 3.4 VALIDATING SOURCES OF MODEL INACCURACIES THROUGH SIMULATION

Due to the scarcity and lack of control of real data, we take advantage of the simulator introduced in Sec. 2.4 to assess whether model inaccuracies originate from community-specific features. We remark that these experiments are carried out on simulated data, generated through a simplified process, and therefore results in this section are meant to undergo further validation in the real world.

First, we investigate the effect of the density of community interactions on the performance of prediction models. We simulate communities with varying amount of interactions between bacteria by controlling the probability of an edge in the interaction matrix. We observe that, as we simulate

denser interactions in the train and test sets, the GNN accuracies stay stable on average (Fig. 7 A). For the MLP, we show two versions. MLP* receives input bacteria in a fixed order. Thus, it can solely rely on positional information, and avoid extracting information from the genome. Consequently, it cannot generalize to unseen bacteria (Fig. 7 B). This MLP* shows the prediction performance that could be obtained by overfitting to each bacterium. The second version, marked MLP, receives bacteria in a shuffled order as input in training and testing. It is forced to extract all information from the genomes, but is evidently unable to make useful predictions.

Next, we test the ability of the models to generalize to unseen keystone bacteria, which could explain the drop in accuracy for certain species in Fig. 6. For that, we increase the edge density for two specific bacteria by increasing the probability of an edge to exist from 0.2 to 0.8 for these nodes only. We simulate communities, exclude each keystone from training, and predict the growth for communities including them as in Sec. 3.3. We perform the same procedure on five non-keystone bacteria for comparison. However, the results do not validate our hypothesis (Fig. 7 B). This implies that GNNs are, in principle, capable of generalizing well to keystone bacteria and that other factors may explain the lack of generalization to AC and its communities in BARANWALCLARK2022. In Appendix B and Supplementary Fig. S10, we additionally study the aspect of scaling the number of bacterial species across all communities, while training on a sufficient number of communities. We find that increasing the diversity of bacteria seen during training helps generalize to unseen species, while maintaining good accuracy for seen ones.

Finally, we explore the impact of community sizes in training versus testing sets. For that, we initially assess whether we can reproduce the decrease in accuracy when generalizing to larger communities with our simulations (see Fig. 3 B). Crucially, while in real data higher-order interactions can drive the drop in accuracy on the test sets, this effect cannot be verified with our simulated data, as it includes only pairwise interactions. In fact, we see a decrease in accuracy when the *size* of the training communities is reduced compared to the test communities (Fig. 7 C). Specifically, models trained on samples with communities of up to 10 bacteria cannot accurately predict communities of 16 to 25 bacteria ($R^2 < 0$). Furthermore, we find that, in simulation, relative abundances are systematically over-estimated in predictions with larger communities. This is likely a consequence of the higher relative abundances in the smaller communities of the training set, indicating a tendency to overfit to training communities. It also suggests that in real data where over- and under-estimations are observed, other factors must influence the lack of generalization. Moreover, we report a trend of higher accuracy when a larger *number* of communities is used for training, while controlling for community size. We investigate this effect in more detail in Appendix B and Supplementary Fig. S9. We show that our approach can scale to more complex microbial networks of up to 200 bacterial species when provided with sufficient samples for training.

Overall, our results suggest that it is crucial to ensure that sufficient data is gathered along three axes: (1) a sufficient number of bacterial species, (2) a sufficient number of community samples, and (3) communities of size similar to target.

## 4 CONCLUSION

Our work sets the stage for the application of GNNs to microbial communities. These models can implicitly learn growth dynamics, and empirically outperform MLPs in terms of accuracy and generalization. Empirically, they outperform MLPs in terms of accuracy and generalization capabilities. Altogether, GNNs hold great potential for further applications. Furthermore, our results show that genomes are sufficient to learn an accurate model that can generalize predictions beyond observed communities. To our knowledge, this is the first attempt at predicting microbial community profiles from genomes directly. Recently, Lam et al. (2020) employed genome-scale metabolic models (GEMs) (van den Berg et al., 2022) adapted for microbial communities (Machado et al., 2018) to predict pairwise bacterial interactions. Hence, a potential next step would be to apply GNNs to such GEMs. Finally, our simulations provide a flexible data generation procedure, which can be used to benchmark models for bacterial growth from genomes. In the future, the simulation can be further improved to account for higher-order interactions and potentially environmental factors. Nonetheless, we hope that its accessibility will encourage the explainable ML community to develop tools to interpret GNN models of bacterial communities. As new properties emerge from microbial communities, scientific discoveries may arise from interactions between our fields.

## REPRODUCIBILITY STATEMENT

We will make our code as well as the trained models available on our project webpage https://sites.google.com/view/microbegnn such that all figures presented in the paper can be reproduced. The real-world datasets used in our work are open-source for which we thank the authors of Friedman et al. (2017) and Baranwal et al. (2022). The datasets can also be found on our webpage. Implementation details and training parameters are detailed in Appendix A.

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

# Appendix

## A    IMPLEMENTATION DETAILS

The datasets contain replicates for some of the settings. To make evaluation simple, we average the results of replicates when included in the validation or test sets. Cross-validation (CV) was performed on 5 train/validation/test data splits with 5 model initialization seeds for hyperparameter tuning (see Supplementary Table S2).

We chose the hyperparameter combination resulting in the lowest validation error. Consequently, we used 2 convolutional layers/message-passing steps with 50 and 100 hidden features for FRIEDMAN2017 and BARANWALCLARK2022 data, respectively. Note that for all architectures, we first have an embedding layer that maps the input genomes to the hidden feature size. For predictions on the test sets, we show each of the 3-models and the average of the 3-models ensemble on Fig. 3 and Fig. 6. Otherwise, predictions on test sets correspond to the average of the 3-models ensemble. As we assume no prior knowledge of bacterial interactions, our graphs are fully connected. For the MPGNN, we used single layer MLPs for the node and edge update functions $g_v$ and $g_e$ (see Eq. 3). These layers were followed by layer normalization and a ReLU activation. For all architectures, output predictions were made with a linear layer followed by the sigmoid function to constrain values into [0, 1]. Note that applying the SoftMax instead of sigmoid did not improve models (Supplementary Table S2). We trained the spatial-convolution GNN with `GraphSAGE` layers implemented in the PyTorch Geometric Python package (Fey & Lenssen, 2019) with mean aggregation. We also apply ReLU activation after each `GraphSAGE` layer and layer normalization is applied before the activation in the final layer.

### A.1    MODEL TRAINING

For all models, the batch size was 16, training samples were shuffled for making batches, and the learning rate was set to 0.005 for the Adam optimizer (Kingma & Ba, 2015).

**Cross-validation on real data**    For cross-validation (Fig. 3 A), data were split into 80/10/10 % train/validation/test sets; five splits were created. We trained models for 500 epochs, with the Adam optimizer to minimize the Mean Squarred Error (MSE); five seeds per model were used. The MSE on the validation set was used to select parameters; we assessed the number of layers, the number of hidden features, whether to train on the average of replicates or each replicate, and whether to apply a SoftMax instead of Sigmoid function after making predictions. Additionally, for the FRIEDMAN2017 dataset, the position of bacteria in the community was shuffled for predictions on the test set for Fig. 3. For the BARANWALCLARK2022 dataset, CV was performed on non-shuffled samples. Models' performances according to parameters are given in Supplementary Table S2.

**Fitting of models on real data**    When no cross-validation was performed, data were split according to community size (Fig. 3 B) or composition (Fig. 6) and no validation set was used. Models were trained for 500 epochs, five seeds were used.

**Fitting of models on simulated data**    We simulated a community of 25 bacteria as described in Sec. 2.4. Samples were created by randomly drawing a subset of bacteria and calculating their relative abundance at equilibrium, also as described in Sec. 2.4. Unless mentioned (Fig. 7 C), 100 samples were generated for training, 10 for validation, and 10 for testing. Models were trained for 1000 epochs; results on test sets are given in Supplementary Table S5.

### A.2    REAL DATASETS

**Friedman2017**    The first set of data was published by Friedman et al. (2017). Experimental data consisted of the relative abundances of bacteria in 2, 3, 7, and 8-bacteria communities at the beginning of the experiment and after 5 days of daily passage, i.e. a fraction of the culture is re-inoculated into fresh growth media. For each mix of bacteria, several initial inoculum ratios were used; 248 samples were performed in duplicates, and 25 samples were not replicated.

Growth curves for mono-cultures are shown in Supplementary Fig. S3, and relative abundances of bacteria in co-cultures are shown in Supplementary Fig. S4 and Supplementary Fig. S5. Given our task to predict stable states of bacterial communities from genomes, we exclude data from mono-cultures and treat mixes started from different inoculum ratios as one sample. Hence, the final dataset consisted of 93 samples with 2 to 15 replicates each. Samples were randomly split in 80/10/10 % train/validation/test sets for cross-validation (CV). We perform experiments in which we exclude 1 bacterium at a time. For those, bacterial communities of 7- and 8-bacteria were excluded from training and testing, so only samples of 2- and 3-bacteria communities were used. For the experiments testing for generalization to bigger communities, training was performed on 2- and 3-bacteria communities and testing on 7- and 8-bacteria communities.

Raw experimental data was kindly provided by Friedman et al. (2017) and is now available on our project website: `https://sites.google.com/view/microbegnn`.

**BaranwalClark2022** The second dataset was published by Baranwal et al. (2022), with certain samples originally produced by Clark et al. (2021). From the initial 1,850 replicates, we removed 258 with records of contamination or "Low Abundance", 39 with an OD600 $\leq 0.1$, and 593 which had more than 0.1 % of non-inoculated bacteria – despite not being recorded as contaminated by authors, resulting in 960 replicates from 459 samples. Each sample was replicated 1 to 9 times.

CV was performed on random 80/10/10 % train/validation/test splits. We carefully looked at the community size representation when excluding each bacterium, and removed samples from the test set if their community size had not been seen during training or only for a few samples (Supplementary Table S1). In particular, we did not attempt to generalize to (i) *Holdemanella biformis* (HB) as only 2- and 26-bacteria communities had been produced with this bacterium, making up a very small test set, and (ii) *Coprococcus comes* (CC), *Eubacterium rectale* (ER), *Roseburia intestinalis* (RI), and *Faecalibacterium praustnitzii* (FP) due to their over-representation in samples, and so the resulting small training sets.

### A.3 ADDITIONAL ARCHITECTURES

In addition to the `MPGNN` and `GraphSAGE` architectures, we also tested improved versions of the Graph Attention (Veličković et al., 2018; Brody et al., 2022) (GATII) and Graph Convolution (Kipf & Welling, 2017; Chen et al., 2020) (GCNv2) architectures. We fitted models on the five CV folds of real data, BARANWALCLARK2022 and FRIEDMAN2017, as for the `MPGNN` and `GraphSAGE` models. The average coefficient of determinations ($R^2$) of the ensemble models fitted on each dataset, with a 95% confidence interval, and calculated on 100 bootstraps of test samples, are reported in the Supplementary Table S3.

## B ADDITIONAL RESULTS

### B.1 SCALABITY OF THE METHOD

We evaluate the scalability of our approach on datasets made of communities of sizes 5-20, 12-25, 25-100, and 50-200 (Supplementary Fig. S9). Note that these communities contain 20, 50, 100 and 200 different bacteria species respectively.

We find that models trained on as few as 50 samples can already generalize in-distribution. Here, in-distribution refers to communities of the same size as seen during training, but with new combinations of the bacterial species in the data. Models trained on larger communities benefit from increasing the number of samples for training. For instance, in the experiment with a set of 200 bacteria and community sizes of 50 to 200 bacteria, we see a jump in generalization performance when we increase the number of samples in the training set from 50 to 250. With only 50 samples, the network doesn't see enough combinations of bacteria in communities to generalize to unseen combinations at test time. However, as we further increase the sample size from 250 to 500, we only see a marginal improvement. This indicates that for a training set containing 200 species, the combinations seen in 250 samples are seemingly adequate for a GNN to generalize.

Given the jump in complexity of communities when increasing community sizes, overfitting with smaller communities and lower generalization for larger ones is expected.

## B.2 DIVERSITY

Finally, we assessed the effect of out-of-distribution bacteria on models' predictions. For that, we fixed (i) a set of 20 baseline bacteria and (ii) a set of 10 bacteria used only in test samples. Training samples consisted of the 20 baseline bacteria, incrementally augmented with more bacteria to create training sets of bacterial diversity, i.e. number of different bacteria, of 20, 50, and 100. The test set comprised communities made of the baseline bacteria plus one test bacterium. We gathered predictions on baseline bacteria, i.e. "seen" during training, and out-of-distribution, i.e. "unseen", and calculated the $R^2$ (goodness of fit) for each type across all test samples. We show results with one unseen bacterium in test communities in the Supplementary Fig. S10. In this case, we see that a higher bacterial diversity seen during training time increases the generalization performance on the unseen bacteria at test time for the MPGNN. Notably, we observe an overall high accuracy of `MPGNN` on seen bacteria despite including an unseen one to the community, indicating strong robustness of the learnt GNN models.

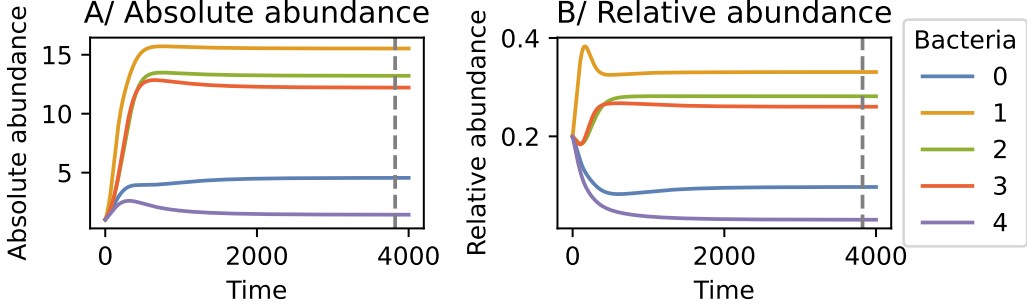

Figure S1: **Growth curves for a simulated bacterial community of 5 members.** The grey dotted line indicates the time point considered for equilibrium. In practice, the relative abundances (B) at this point would be used as the target for fitting models.

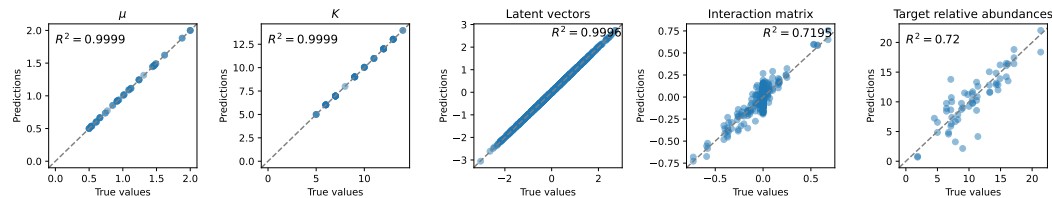

Figure S2: **Recovery of growth parameters from genomes.** Most parameters can be reconstructed to high accuracy. Approximations errors compound in the reconstructed interaction matrix, but interaction coefficients can still be reconstructed with a coefficient of determination $R^2 > 0.7$.

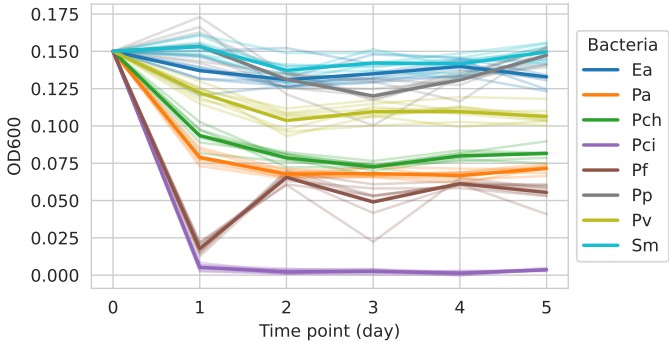

Figure S3: **Growth curves of bacteria in mono-cultures from Friedman et al. (2017).** Thin lines correspond to replicates and the thick line corresponds to the average at each time point across replicates; cultures were passed into fresh medium daily.

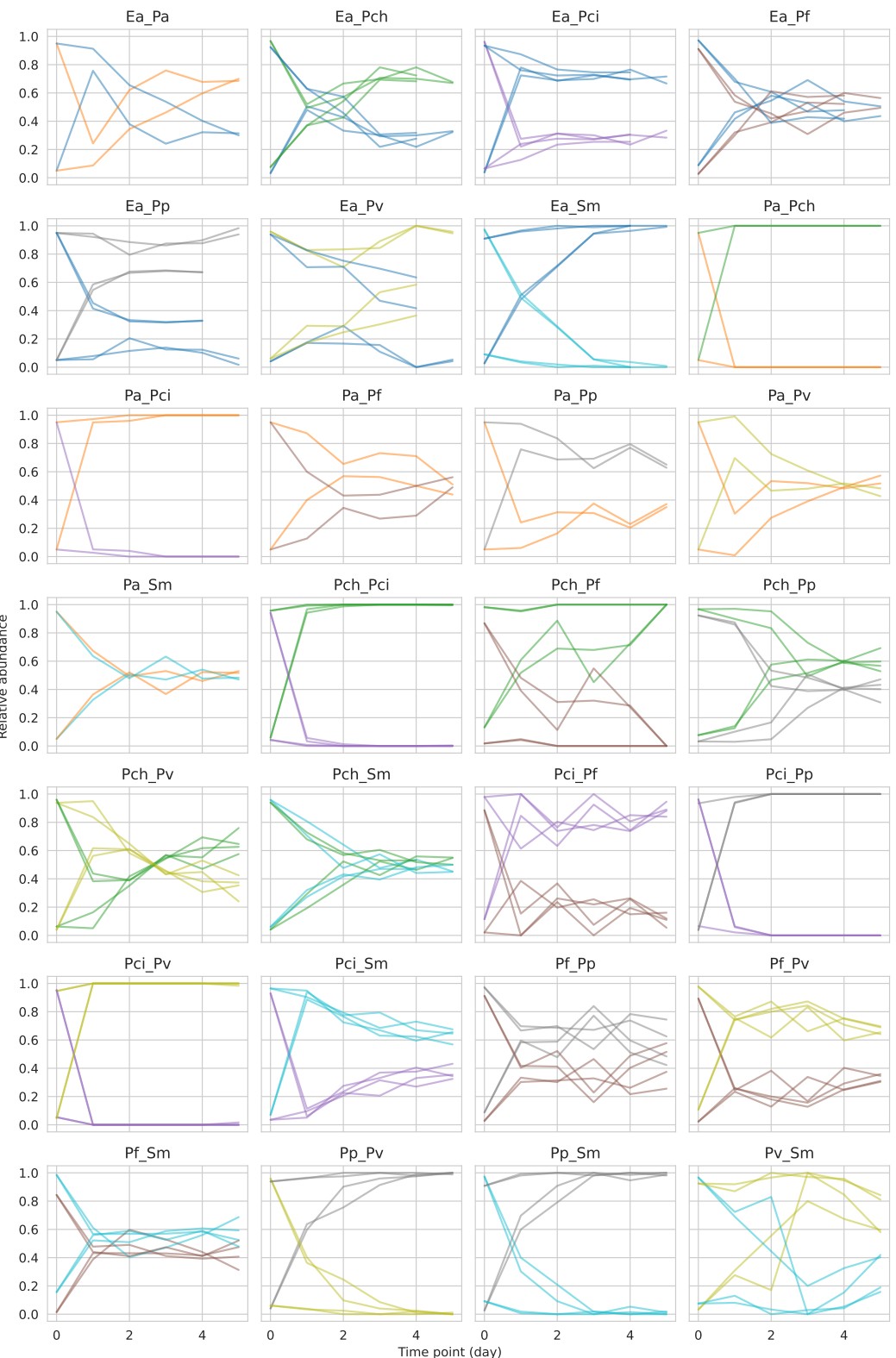

Figure S4: **Relative abundances of bacteria in co-cultures from Friedman et al. (2017).** Each line corresponds to a replicate; cultures were started with 0.95 / 0.05 relative abundances of each bacterium.

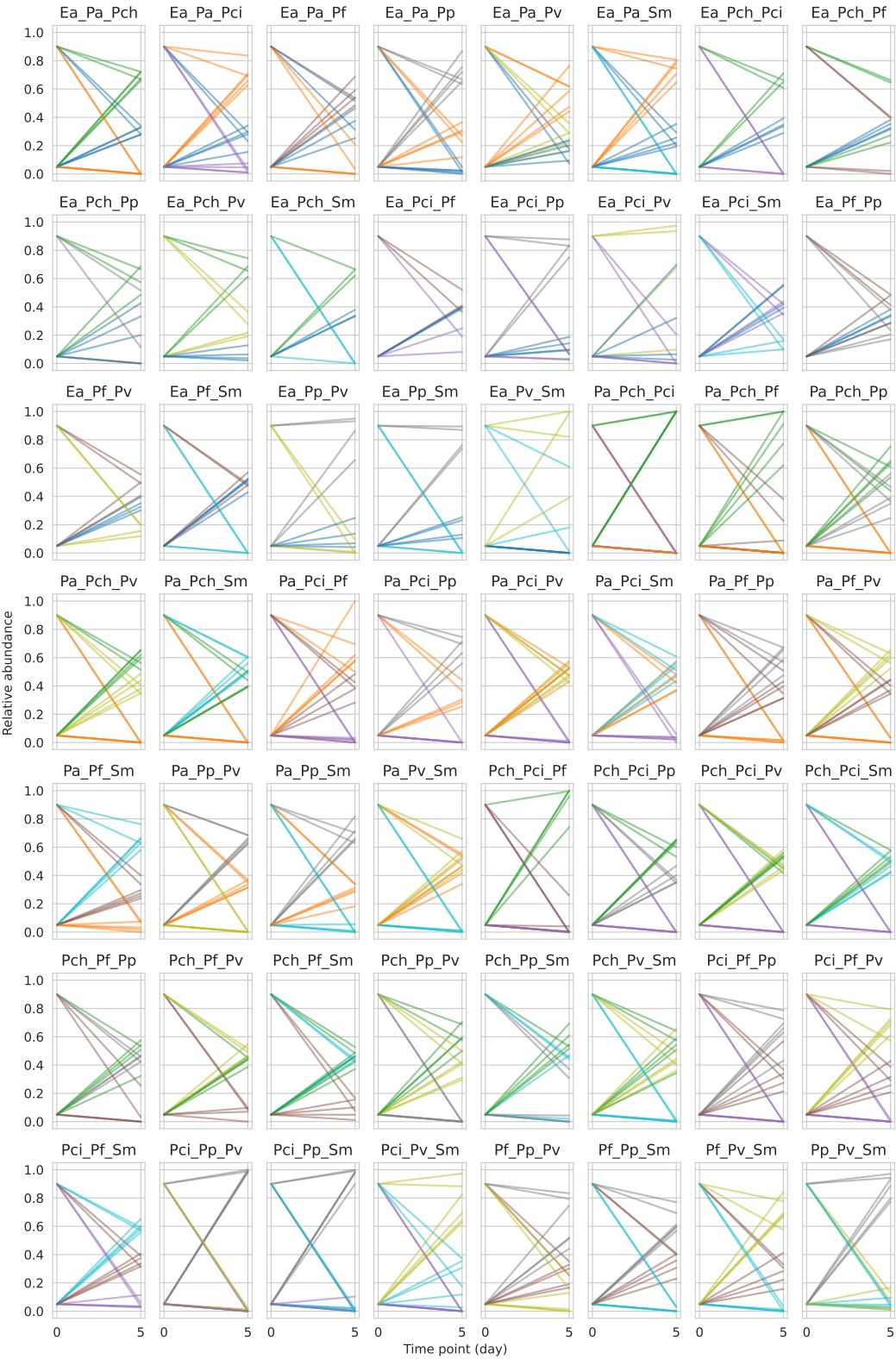

Figure S5: **Relative abundances of bacteria in 3-bacteria cultures from** Friedman et al. (2017). Each line corresponds to a replicate; cultures were started with different relative abundance ratios.

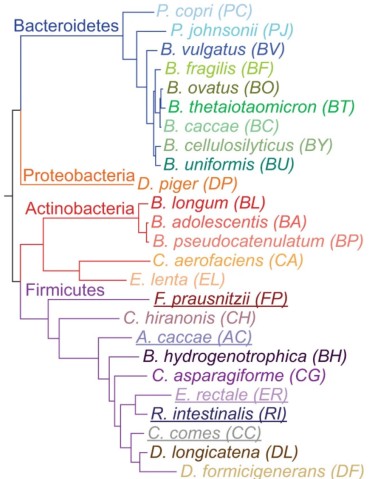

Figure S6: **Phylogenetic tree of bacteria used in the BaranwalClark2022 data set (from Clark et al. (2021)).** Tree branches are colored by phylum and underlined bacteria are butyrate producers.

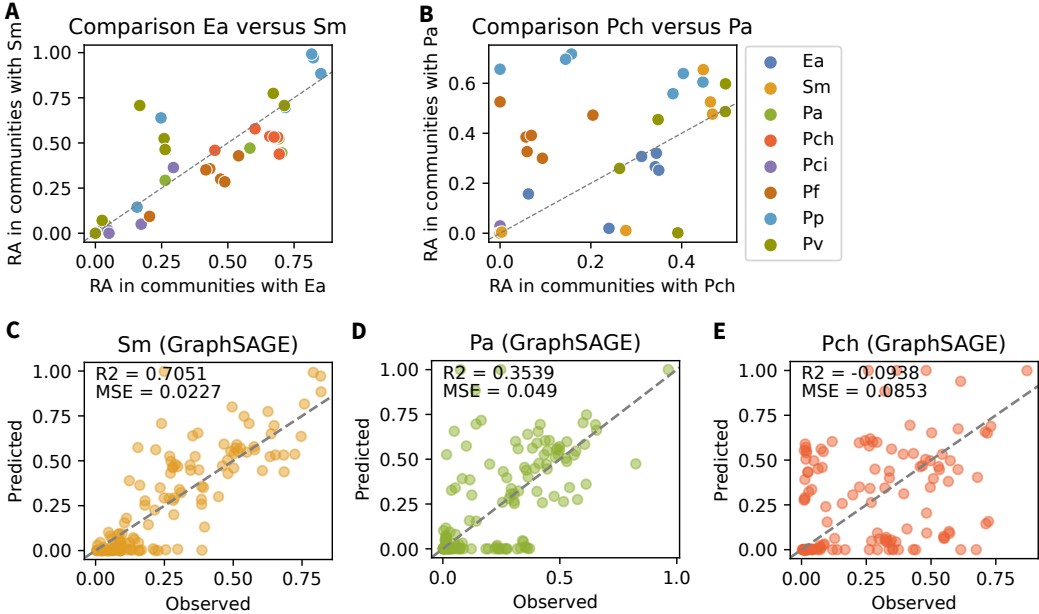

Figure S7: **Comparison of the effect of two bacteria on the relative abundances of others in 2- and 3-bacteria cultures, in the Friedman2017 data set.** For each bacterium on the x and y-axis, communities were matched by co-partners, and the average relative abundances of the bacteria are shown, colored by bacteria. Suppose the x- and y-axis bacteria had similar interaction effects with their partners in the matched communities. In that case, the relative abundances of the other bacteria should be similar and so, close to the $x = y$ grey dotted line. A/ The two non-*Pseudomonas* resulted in resembling communities when grown with *Pseudomonas*, with a mean squared distance between relative abundances in matching communities of 0.0078. B/ Despite being the phylogenetically closest strains, Pch and Pa resulted in different communities C-D/ Predicted versus observed relative abundances when generalizing to Sm, Pa, and Pch. The average relative abundances across replicates are shown for the observed values.

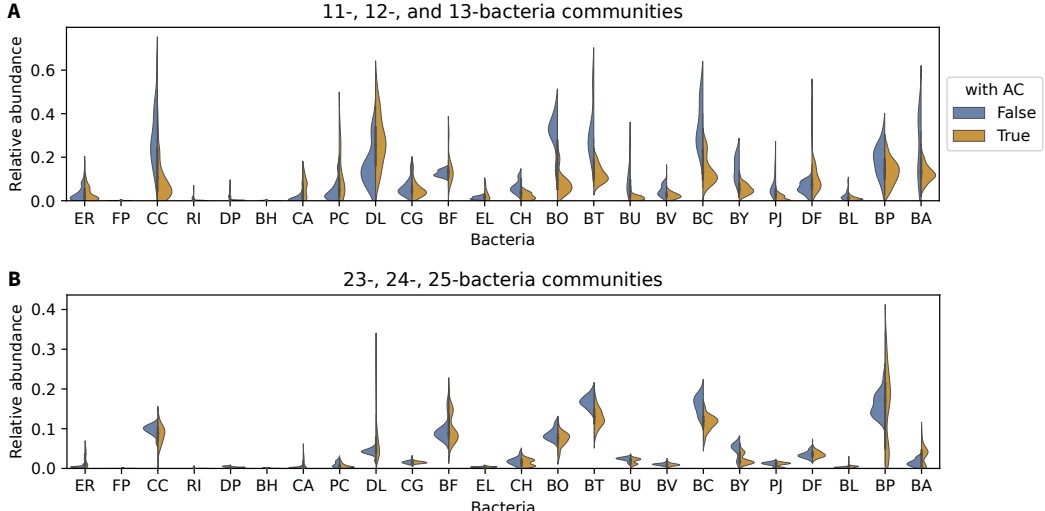

Figure S8: **Comparison of relative abundances of bacteria when grown in communities containing or not *Anaerostipes caccae* (AC).** Due to the scale of relative abundances according to the size of the community, we show as examples results for communities of A/ 11 to 13-bacteria and B/ 22 to 25-bacteria.

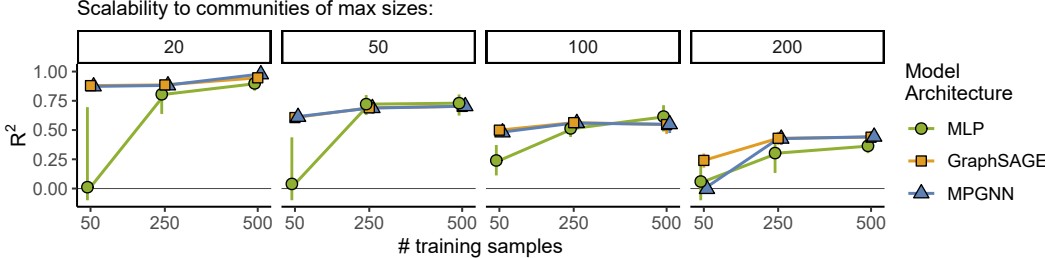

Figure S9: **Scalability of GNNs to larger communities.** We simulate sets of $n = 20, 50, 100,$ and 200 bacteria, and draw communities of size $n/4$ to $0.9 * n$ for training and testing. All bateria in the test-sets have been observed. Models are trained on increasing number of samples, showing an increase in accuracy throughout. The MLP gets the bacteria in a fixed order. For this reason, we observe good performance with larger training set sizes.

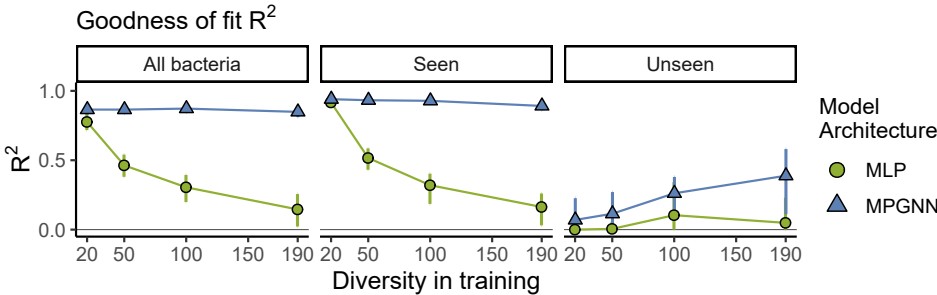

Figure S10: **Generalization of GNNs to unseen bacteria improves with higher diversity seen during training.** We simulate sets of $n = 20, 50$ and 200 bacteria, and draw communities of size $n/4$ to $0.9 * n$ for training. At test time, we introduce a new bacteria that has not been seen before and test the generalization capabilities of our models on communities including this new bacterium that was not seen at training time. The test sets are shared across models, and so all bacteria, except the new one, are from the smallest set of 20 bacteria. We use 10,000 samples for training all models.

Table S1: Species and community sizes excluded from the test sets for results in 3.3.

| Species | Community sizes excluded from test set |
|---------|----------------------------------------|
| CH | 24-26 |
| BT | 24-26 |
| DP | 23-26 |
| BL | 22-26 |
| BH | 22-26 |
| CG | 22-26 |
| EL | 22-26 |
| BF | 22-26 |
| PJ | 22-26 |
| BY | 22-26 |
| BA | 18-26 |
| DL | 18-26 |
| BP | 18-26 |
| CA | 18-26 |
| BV | 18-26 |
| BC | 18-26 |
| PC | 2-4, 17-26 |
| BU | 2-4, 17-26 |
| BO | 2-10, 17-26 |
| DF | 16-26 |

---

[1]BC: BaranwalClark2022, F: Friedman2017
[2]BC: BaranwalClark2022, F: Friedman2017

Table S2: **Mean Squared Error of models on validation sets after 500 epochs of training.** #conv refers to the number of convolution layers stacked for GraphSAGE and the number of message-passing steps for MPGNN, both corresponding to the depth of information propagation in the graph.

| Dataset[1] | Architecture | # conv | n hidden features | MSE validation set | Comment |
|---|---|---|---|---|---|
| F | MPGNN | 2 | 50 | 0.013607 | |
| F | GraphSAGE | 2 | 200 | 0.014383 | |
| F | GraphSAGE | 2 | 50 | 0.014627 | |
| F | MPGNN | 2 | 200 | 0.014736 | |
| F | GraphSAGE | 3 | 50 | 0.014987 | |
| F | MLP | 2 | 100 | 0.015226 | no permutation |
| F | MPGNN | 2 | 100 | 0.015554 | |
| F | GraphSAGE | 1 | 100 | 0.01597 | |
| F | MPGNN | 3 | 50 | 0.016013 | |
| F | GraphSAGE | 2 | 100 | 0.016515 | |
| F | MPGNN | 1 | 100 | 0.035642 | |
| F | MLP | 2 | 100 | 0.051399 | |
| BC | GraphSAGE | 2 | 200 | 0.006484 | |
| BC | MPGNN | 2 | 100 | 0.006641 | train average replicates |
| BC | GraphSAGE | 2 | 100 | 0.006683 | |
| BC | MPGNN | 2 | 100 | 0.006821 | |
| BC | GraphSAGE | 2 | 50 | 0.006881 | |
| BC | MPGNN | 2 | 200 | 0.006897 | |
| BC | GraphSAGE | 3 | 100 | 0.006986 | |
| BC | MPGNN | 2 | 50 | 0.007218 | |
| BC | GraphSAGE | 2 | 100 | 0.007532 | train average replicates |
| BC | GraphSAGE | 2 | 100 | 0.007538 | SoftMax instead of Sigmoid |
| BC | GraphSAGE | 1 | 100 | 0.007584 | |
| BC | MPGNN | 1 | 100 | 0.015077 | |
| BC | MPGNN | 3 | 100 | 0.023026 | |
| BC | MPGNN | 2 | 100 | 0.811864 | SoftMax instead of Sigmoid |

Table S3: **Mean Squared Error of models on validation sets after 500 epochs of training and coefficient of determination $R^2$ on test sets, with a 95% confidence interval.** #conv refers to the number of convolution layers, corresponding to the depth of information propagation in the graph.

| Dataset[2] | Architecture | # conv | n hidden features | MSE validation set | $R^2$ test set |
|---|---|---|---|---|---|
| F | GCNII | 1 | 50 | 0.0778 (0.0676, 0.0879) | 0.0618 (-0.1410, 0.2822) |
| F | GATv2 | 1 | 50 | 0.0724 (0.0642, 0.0805) | 0.0279 (-0.0808, 0.1266) |
| BC | GATv2 | 1 | 50 | 0.0468 (0.0379, 0.0556) | 0.3569 (0.2455, 0.4711) |
| BC | GATv2 | 1 | 100 | 0.0478 (0.0399, 0.0556) | 0.1034 (0.0075, 0.1914) |
| BC | GATv2 | 2 | 50 | 0.0539 (0.0464, 0.0613) | -0.1621 (-0.2635, -0.0975) |
| BC | GATv2 | 2 | 100 | 0.0550 (0.0469, 0.0632) | -0.1305 (-0.1993, -0.0840) |
| BC | GCNII | 1 | 50 | 0.0405 (0.0327, 0.0483) | 0.5059 (0.2705, 0.6959) |
| BC | GCNII | 1 | 100 | 0.0435 (0.0311, 0.0558) | 0.2080 (-0.1771, 0.4842) |
| BC | GCNII | 2 | 50 | 0.0219 (0.0173, 0.0264) | 0.6793 (0.4865, 0.8143) |
| BC | GCNII | 2 | 100 | 0.0235 (0.0187, 0.0283) | 0.7090 (0.5498, 0.8277) |

Table S4: Bacterial strains in experimental data, their designation in the article, and the genomes used to fit models.

| Data[3] | Bacterial strain | Designation | Substitute genome | Database |
|---|---|---|---|---|
| BC | Anaerostipes caccae L1-92 | AC | | NCBI |
| BC | Bacteroides cellulosilyticus CRE21 | BY | | NCBI |
| BC | Bacteroides uniformis VPI 0061 | BU | | NCBI |
| BC | Bifidobacterium adolescentis E194a (Variant a) | BA | | ATCC |
| BC | Bifidobacterium longum subs. infantis S12 | BL | | NCBI |
| BC | Bifidobacterium pseudocatenulatum B1279 | BP | | NCBI |
| BC | Blautia hydrogenotrophica S5a33 | BH | | NCBI |
| BC | Clostridium asparagiforme N6 | CG | | NCBI |
| BC | Clostridium hiranonis T0-931 | CH | | NCBI |
| BC | Collinsella aerofaciens VPI 1003 | CA | | NCBI |
| BC | Desulfovibrio piger VPI C3-23 | DP | | NCBI |
| BC | Dorea formicigenerans VPI C8-13 | DF | | NCBI |
| BC | Dorea longicatena 111–35 | DL | | NCBI |
| BC | Eggerthella lenta 1899 B | EL | | NCBI |
| BC | Bacteroides caccae VPI 3452 A | BC | Bacteroides caccae CL03T12C61 | NCBI |
| BC | Bacteroides fragilis EN-2 | BF | | NCBI |
| BC | Bacteroides ovatus NCTC 11153 | BO | | NCBI |
| BC | Bacteroides thetaiotaomicron VPI 5482 | BT | | NCBI |
| BC | Bacteroides vulgatus NCTC 11154 | BV | | ATCC |
| BC | Coprococcus comes VPI CI-38 | CC | | NCBI |
| BC | Eubacterium rectale VPI 0990 | ER | | ATCC |
| BC | Faecalibacterium prausnitzii A2-165 | FP | | NCBI |
| BC | Parabacteroides johnsonii M-165 | PJ | | NCBI |
| BC | Prevotella copri CB7 | PC | | NCBI |
| BC | Roseburia intestinalis L1-82 | RI | | NCBI |
| BC | Holdemanella biformis DSM 3989 | HB | | NCBI |
| F | Enterobacter aerogenes ATCC 13048 | Ea | | NCBI |
| F | Pseudomonas aurantiaca ATCC 33663 | Pa | Pseudomonas chlororaphis strain qlu-1 | NCBI |
| F | Pseudomonas chlororaphis ATCC 9446 | Pch | | NCBI |
| F | Pseudomonas citronellolis ATCC 13674 | Pci | Pseudomonas citronellolis strain P3B5 | NCBI |
| F | Pseudomonas fluorescens ATCC 13525 | Pf | | NCBI |
| F | Pseudomonas putida ATCC 12633 | PP | | NCBI |
| F | Pseudomonas veronii ATCC 700474 | PV | Pseudomonas veronii strain ASM202832 | NCBI |
| F | Serratia marcescens ATCC 13880 | Sm | | NCBI |

---

[3]BC: BaranwalClark2022, F: Friedman2017

Table S5: Mean Squared Error on validation set and coefficient of determination $R^2$ on test sets of models fitted on simulated data after 250 epochs of training. MLP and MLP* are the same models, hence they have the same MSE on validation set; MLP received shuffled input bacteria for the test set (similar to GNNs) while MLP* did not (optimal conditions).

| Model | Edge density | MSE validation set | $R^2$ test set |
|---|---|---|---|
| MLP* | 0.1 | 0.000202 | 0.8364 |
| MLP* | 0.4 | 0.000563 | 0.7798 |
| MLP* | 0.6 | 0.000566 | 0.7173 |
| MLP | 0.1 | 0.001205 | -0.1911 |
| MLP | 0.4 | 0.003315 | -0.2980 |
| MLP | 0.6 | 0.003471 | -0.4157 |
| GraphSAGE | 0.1 | 0.001592 | 0.4749 |
| GraphSAGE | 0.4 | 0.001924 | 0.4034 |
| GraphSAGE | 0.6 | 0.002191 | 0.4184 |
| MPGNN | 0.1 | 0.001612 | 0.4324 |
| MPGNN | 0.4 | 0.001799 | 0.6016 |
| MPGNN | 0.6 | 0.001882 | 0.4173 |
| **Excluded bacteria** | | | |
| MLP* | key | 0.000897 | -7.3473 |
| MLP* | random | 0.000926 | -8.5531 |
| MLP | key | 0.011816 | -0.1689 |
| MLP | random | 0.013400 | -0.1162 |
| GraphSAGE | key | 0.001101 | 0.4900 |
| GraphSAGE | random | 0.0011308 | 0.4465 |
| MPGNN | key | 0.000576 | 0.6438 |
| MPGNN | random | 0.000756 | 0.5594 |
| **Max training community size / training sample size** | | | |
| MLP* | 10 / 100 | 0.002040 | -1.0551 |
| MLP* | 10 / 200 | 0.000906 | -0.0141 |
| MLP* | 15 / 50 | 0.000700 | 0.4234 |
| MLP* | 15 / 100 | 0.000306 | 0.7998 |
| MLP* | 15 / 200 | 0.000123 | 0.8959 |
| MLP* | 25 / 100 | 0.00324 | 0.8147 |
| MLP | 10 / 100 | 0.002953 | -1.3241 |
| MLP | 10 / 200 | 0.002516 | -1.3176 |
| MLP | 15 / 50 | 0.001244 | -0.0740 |
| MLP | 15 / 100 | 0.001263 | 0.0011 |
| MLP | 15 / 200 | 0.001278 | 0.0404 |
| MLP | 25 / 100 | 0.001668 | -0.066 |
| GraphSAGE | 10 / 100 | 0.002823 | -1.7475 |
| GraphSAGE | 10 / 200 | 0.001779 | -0.8447 |
| GraphSAGE | 15 / 50 | 0.001227 | -0.3847 |
| GraphSAGE | 15 / 100 | 0.00100 | 0.1108 |
| GraphSAGE | 15 / 200 | 0.000789 | 0.3243 |
| GraphSAGE | 25 / 100 | 0.000760 | 0.4360 |
| MPGNN | 10 / 100 | 0.002595 | -1.5964 |
| MPGNN | 10 / 200 | 0.001689 | -0.8451 |
| MPGNN | 15 / 50 | 0.001529 | -0.5713 |
| MPGNN | 15 / 100 | 0.000996 | 0.1456 |
| MPGNN | 15 / 200 | 0.000503 | 0.6597 |
| MPGNN | 25 / 100 | 0.000764 | 0.4579 |

