# OpenReview forum: "Modelling Microbial Communities with Graph Neural Networks"
_ICLR.cc/2024/Conference — Submitted to ICLR 2024_

### Official Review · Reviewer_U7B8 · 2023-10-23

**Soundness:** 3 good
**Presentation:** 4 excellent
**Contribution:** 3 good
**Rating:** 6
**Confidence:** 3

**Summary:**

The study focuses on understanding the interactions between microorganisms, which is of significant importance in both medical and environmental contexts. The authors introduce a novel approach by modeling bacterial communities using graph neural networks (GNNs) directly from the genomes of the bacteria. The inherent properties of GNNs, such as permutation invariance, allow them to effectively capture the relationships within the bacterial set, thus offering combinatorial generalization.

**Strengths:**

- Novel problem setup and the first use of GNN to tackle this problem.
- The use of GNN matches with the data well since it is modeling a dynamic system.
- Very interesting set of experiments and they are extensive.
- The presentation is nice and clear.
- Nice simulation data construction and results.

**Weaknesses:**

- Methodological novelty is limited since it is basically fitting a GNN on a bacterial community graph. This is not to say the novelty of the paper is limited. Since I do believe it is tackling an interesting new problem with impact. I would suggest the authors consider a journal paper instead.

**Questions:**

Where are the circles for fig3A (models not on permuted data)? Why only select some of the combinations and not showing all of them? It would also be great if the authors could compare with standard practice of this task instead of just comparing with GraphSAGE. For example, by fitting the mechanistic model.

Have the authors experimented with other GNN models? Since graphsage is only one instantiation and there are many recent ones with more expressive powers.

Modeling the dynamics sounds interesting. Could the authors also use GNN in an iterative way to model the dynamics? For example, using ideas from this paper: http://proceedings.mlr.press/v119/sanchez-gonzalez20a.html

---

> ### Author Response · Authors · 2023-11-22
> **Response to Reviewer U7B8**
>
> Thank you for your positive evaluation of our work and feedback.
>
> **Fig3A**
> They are now shown as the average performance of the ensemble model across the 5 cross-validation sets is shown together with the 95% confidence interval. We have homogenized the figures now to show the average and the 95% confidence interval.
>
> **Combinations**
> We have now included the MLP architecture in all experiments for more homogeneity in the manuscript, we hope this clarifies the reviewer's concern.
>
> **Other architectures**
> As per the reviewer’s suggestion, we have now tested the improved graph attention architecture (GATv2) [1] (original version GAT in [2]), and the improved graph convolution from [3] (GCNII), (original version GCN in [4]). Unfortunately, they failed to fit the Friedman2017data (see Table below). We added results in Appendix A and Supplementary Table S3.
>
> **Table:** Accuracy of models implemented with the GATv2 [2] and GCNII [4] architectures on real microbial community data. For each dataset, five seeds per model were used and models fitted on five cross-validation folds. The average coefficient of determination ($R^2$) of the ensemble models, with a 95% confidence interval, was calculated on 100 bootstraps of test samples.
>
> | Dataset | Architecture | Test average $R^2$ | 95 % confidence interval |
> |-----------|------------------|---------------------------|---------------------------------|
> | Friedman2017 | GCNII | 0.0618 | [-0.1410, 0.2822] |
> | Friedman2017 | GATv2 | 0.0279 | [-0.0808, 0.1266] |
> | Friedman2017 | MPGNN | 0.808809  | [0.677719, 0.894381] |
> | BaranwalClark2022 | GCNII | 0.7090 | [0.5498, 0.827654] |
> | BaranwalClark2022 | GATv2 | 0.3568 | [0.2455, 0.4711] |
> | BaranwalClark2022 | MPGNN | 0.762534 | [0.665613, 0.844525] |
>
> These architectures do not seem to be particularly suitable for our task, as even with hyperparameter optimization, we could not get good performances on both datasets.
>
> **Dynamics**
>
> We agree that explicitly predicting growth dynamics may be ideal for apprehending bacterial community assembly and interactions. In practice, the GNN models we propose can be repurposed for this task, and densely predict the state of the community at arbitrary timesteps. It is also possible to enhance the model by using e.g. recurrent GNN. A challenge is that datasets usually contain one time point per sample due to time and resource costs. Therefore, while this approach would be possible for a few datasets, e.g., as in [5-6], it could not be extended to more.
>
> ----
>
> **References**
>
> [1] Brody, Shaked, Uri Alon, and Eran Yahav. "How attentive are graph attention networks?." arXiv preprint arXiv:2105.14491 (2021).
>
> [2] Veličković, Petar, et al. "Graph attention networks." arXiv preprint arXiv:1710.10903 (2017).
>
> [3] Chen, Ming, et al. "Simple and deep graph convolutional networks." International conference on machine learning. PMLR, 2020.
>
> [4] Kipf, Thomas N., and Max Welling. "Semi-supervised classification with graph convolutional networks." arXiv preprint arXiv:1609.02907 (2016).
>
> [5] Michel-Mata, Sebastian, et al. "Predicting microbiome compositions from species assemblages through deep learning. iMeta 1: e3." (2022).
>
> [6] Baranwal, Mayank, et al. "Recurrent neural networks enable design of multifunctional synthetic human gut microbiome dynamics." Elife 11 (2022): e73870.

---

### Official Review · Reviewer_aYM1 · 2023-10-29

**Soundness:** 3 good
**Presentation:** 2 fair
**Contribution:** 1 poor
**Rating:** 6
**Confidence:** 3

**Summary:**

The paper looks at modeling bacterial communities and their interactions using graph neural networks (GNNs). They rely on two open datasets, total n = 552 samples. The authors have downloaded genomes for the bacteria that was converted to growth encodings. To address the issue with limited data the authors also used a simulator based on the Lotka-Volterra model. They compare three different models, MLP as the standard, GNNs and MPGNN. Using GNN/MPGNN the authors were able to model but the models were sensitive to variations and generalizing to larger systems was poor. Models were better than MLP but only marginally.

**Strengths:**

I found the paper interesting and I think the authors are correct that a better modeling of bacteria would open up a much better understanding of a wide range of fields. Key strengths:
* The authors' comparative approach between models is commendable.
* The paper addresses a clinically relevant topic, shedding light on bacterial interactions.
* The authors' transparency regarding the challenges in scaling the mod

**Weaknesses:**

While I enjoyed reading something on the outskirts of my experience, although I have grown my own tuberculosis communities in the early days of my research, I struggle with some of the basic premises:
* Motivation & Context: The paper's motivation needs clearer alignment with real-world applications. The authors cite that understanding these communities is essential for gut, industry and space but I find the step from this paper to extrapolating to gut seems huge. The largest studied communities are 26 and this needs to be put in context with the other fields, citing Wikipedia “1010 to 1011 cells per gram of intestinal content” seems far off from the estimated single colonies. The types of bacteria should also be matched with the environment that you aim to generalize for.
* Sample Size & DNA Inclusion: I'm concerned about the limited independent samples, especially in combination with the attempt to include DNA. Making sense of DNA has proven much more difficult than thought of in the beginning and I’m not convinced that the addition made sense. Adding it to the paper risks of overfitting the data even more. I wonder if the field wouldn’t benefit more from going from 500 samples to 1-2000 more than this paper. My experience with building models on this type of data is that they are frustratingly brittle due to the lack of data.
* Clarity & Explanation: Coming from medicin to ML is always a challenge. It would be helpful if the paper could provide clearer explanations for terms and metrics, especially for readers transitioning from medical backgrounds. E.g. keystone bacteria are not explained, good vs acceptable R2 is unclear to the reader (I can’t even find clearly how is this calculated, despite looking in appendix A which I should not have to for the main outcome), I assume that R2 is highly dependent on the underlying complexity, also the datasets have completely different bacteria suggesting that their purpose was different but this is unclear to me despite reading it several times.
* Simulation Impact: The paper should provide a clearer explanation of the effect of simulated colonies on the models' stability.
Regarding the conclusion I’m a little confused as to why it doesn’t recommend including more data. I believe the authors have devoted significant time to this paper and before we put others down this path, perhaps we should wait for more data or do the authors truly feel that GEMs will be the solution?

**Questions:**

See weaknesses.

My main question is if it is true that the lack of data was your biggest challenge? And if so I would like to have it clearly stated so that others may look for additional data sources or make their own datasets available before we dive into new models.

---

> ### Author Response · Authors · 2023-11-22
> **Response to Reviewer aYM1**
>
> Thank you very much for your feedback. Hopefully, this work will also be useful to investigate microbial communities within which *Mycobacterium tuberculosis* grows!
>
> **Motivation & Context**
> We refer the reviewer to the common response: we have added an experiment on simulated data to evaluate the scalability of the method. Regarding the type of bacteria and environment, we do not aim for particular microbiota: Friedman2017 showcases soil bacteria and BaranwalClark2022, gut bacteria. In general, our framework could be applied to any microbiota.
>
> **Sample Size & DNA Inclusion**
> Here, the inclusion of DNA allows us to generalize by its universality, as we treat it as a global feature space. It is the only set of features we learn from and, as shown in the results, enables predicting unseen bacteria to a certain extent.
>
> We agree that the lack of data entails good performance, as shown in the new experiments in Appendix B and Supplementary Fig. S9. Our new results clearly highlight that, especially for large, complex communities, models benefit from more samples at training time. They also demonstrate that our approach can scale to large communities of up to 200 bacteria. Finally, we also show in Appendix B and Supplementary Fig. S10, that given a large number of samples, GNNs’ performance on unseen bacteria increases with the diversity of bacteria seen during training. Hence, we believe our method will become increasingly useful with larger-scale datasets.
>
> **Clarity & Explanation**
> - $R^2$: We have now detailed $R^2$ in the Methods of the main text. $R^2$ measures the deviation from a given model to a random one that would predict an average value. Hence, it is independent of the underlying complexity.
> - Datasets: please, see the common response.
> - Keystone bacteria: we have changed the wording in the text. A keystone bacterium is a bacterium that drives community assembly; they are identified from data showing that their presence or removal dramatically affects the community composition [1].
>
> **Simulation Impact & Question**
>
> - Firstly, we have improved clarity on the fact that models were fitted to datasets independently -- please, see the common response.
>
> - The simulations are intended to study the properties of the prediction pipeline. We have now developed this further by investigating scaling and diversity aspects. We added this to Appendix B and discussed it in Section 3.4; see also answer to **Sample Size & DNA Inclusion.**
>
> - We do not intend to suggest that more data is not beneficial: our experiments on simulated data show an increase in model accuracy when trained on more communities sampled from a given set of bacterial species, as discussed in Appendix B.1 and shown in Supplementary Fig. S9. In this particular setting, we observe that accuracy plateaus once a sufficient number of communities is observed.
> Given the jump in complexity of communities when increasing community sizes (in this case the number of bacteria species in the network), we see an overall decrease in performance as overfitting with smaller communities and lower generalization for larger ones is observed.
> When it comes to generalizing to unseen bacteria, we also argue that increasing the number of samples improves predictions to a certain extent but is limited. However, increasing the diversity of bacteria seen during training enables further improving the accuracy as shown in Supplementary Fig. S10.
>
> - Our opinion is that we, as bioinformaticians and computer scientists, should tackle biological challenges with the data that is currently available. While efforts are made in the lab to enhance datasets with high-throughput methods (e.g., sample miniaturization and automation of data acquisition), we make progress on the data analysis side. The datasets that will appear on this topic soon, will still not be big enough to satisfy large-scale deep learning machinery, so we believe our method is timely and at the right scale.
>
> ----
>
> **References**
>
> [1] Banerjee, Samiran, Klaus Schlaeppi, and Marcel GA van der Heijden. "Keystone taxa as drivers of microbiome structure and functioning." Nature Reviews Microbiology 16.9 (2018): 567-576.

---

### Official Review · Reviewer_5GQX · 2023-10-30

**Soundness:** 2 fair
**Presentation:** 2 fair
**Contribution:** 2 fair
**Rating:** 3
**Confidence:** 3

**Summary:**

The paper tested the idea of using MPGNN or GraphSAGE to learn generalizable microbial community steady-state dynamics. The proposed models were tested on  simulated and previous publicly available microbial datasets and compared with the MLP-based implementation to show the effectiveness, with the discussions on the generalizability of GNN-based implementations.

**Strengths:**

The presented comparison results with MLP-based implementation demonstrates the potential of GNN-based implementations to model microbial community dynamics.

**Weaknesses:**

1. The methodological contribution is limited as the presented work is mostly implementing GNNs for microbial steady state predictions.

2. The main core of the paper is based on the assumption that if there is a steady state solution to the dynamics of bacterial species, then that steady state can be predicted using the genome data of the species in the system. This is a reasonable assumption to make. However the fact that this method works only for steady state solutions needs to be emphasized. Indeed in the GLV setting in the famous example of foxes and rabbits, there could be steady state and oscillatory solutions even though the participating genomes are foxes and rabbits in both the cases. It might also be a good idea to highlight why authors expect to find (or not) only steady state solutions in systems involving microorganisms such as bacteria. This will add more strength to the paper.

3. For simulations, the GLV equations along with initial conditions and parameters $\mu_i, K_i, a_{i,j}$ drawn from different probability density functions are used to generate data. Did all such simulations lead to steady state solutions? Were any simulations that did not lead to steady state solutions discarded? Do the authors also have any comments on the frequency of steady state solutions when random parameters are used?

4. Given the GLV equations, the steady state solutions can be found by solving a system of $|S|$ linear algebraic equations:
    \begin{equation}
        \sum_{j=1}^{|S|}a_{i,j}n_j = K_i.
    \end{equation}
The steady state is entirely determined by the parameters $a_{i,j}$ and $K_i$. The authors use a vector composed of $[\mu_i, K_i, \nu_i^s, \nu_i^r, random]$ (where $a_{i,j} \approx \nu_i^s. \nu_j^r$) to simulate the genome data in their simulation. It would be a good idea to highlight that within the simulated genome vector only the components $[K_i, \nu_i^s, \nu_i^r]$ determine the steady state solution.

5. The parameter $a_{i,j}$ (broken into two vectors $\nu_i^s, \nu_j^r$ to simulate the genome) contains information on the pairwise interaction between different species. On the other hand, the information in a genome is completely intrinsic to a particular species. The authors should square these two facts.

6. A proper simulation would entail simulation of the genome data. The genome data typically do not include information on interaction between species. But for simulations, the interaction matrix was used to derive $\nu$ vectors. The claim of interpretability seems to be questionable.

7. The authors appear to be confused on equivariance and invariance. The permutation invariance justification for using graph neural networks is confusing. For example, GLV models are widely used to model the dynamics of microorganisms. But the GLV model is not permutation invariant. The authors stated "\textit{When shuffling the order of bacteria within the train and test communities, the accuracy of MLPs drops significantly, clearly showing that the dynamics learned by MLPs are not invariant to permutations...}" It is to be expected that shuffling the data will lead to reduction in performance of MLP based models. But as long as all the training and testing is done with a particular order of species, it should not matter.

8. The authors need to provide details on how the node (genome) attributes were obtained, especially $\nu$'s, as in real-world data, the ground-truth interaction $a_{ij}$ is not available.

**Questions:**

1. How the nodes, edges and their associated attributes/features were constructed, especially based on the real-world data?

2. How scalable is the GNN-based implementation with respect to the number of microbial species?

---

> ### Author Response · Authors · 2023-11-22
> **Response to Reviewer 5GQX**
>
> Thank you very much for thoroughly evaluating our work, especially on the simulation section. Your comments are much appreciated and highlight points we must clarify in the manuscript. Below, we answer the weakness points (W#) and questions (Q#).
>
> **W1**
> The vast majority of datasets consist of a unique time point per sample. Hence, there is a trade-off between waiting for more growth dynamics and utilizing the data we already have. While efforts are made in the lab to enhance datasets with high-throughput methods (e.g., sample miniaturization and automation of data acquisition), progress on the data analysis side may also be made to fit the data type better. We, as well, would prefer working with growth curves, but these datasets are simply too few and showcase too small communities. Hence, we propose an alternative to focus on steady states, with more datasets and applications. Also, microbiota can be hypothesized to be in pseudo-stable states, as they generally show low variability across time except for big environmental perturbations [1].
>
> **W2-3**
> You are entirely right: not all simulated communities necessarily lead to a steady state; we would discard suicidal and non-stable communities, i.e. keep communities if $\sum_{i \in |S|} n_i > 0$ and $\frac{dn_i}{dt} < 10^{-3} \; \forall i \in |S|$, with $|S|$ the number of bacteria in the community. In practice, for our set of parameters, all simulated communities were already stable, hence we did not need to resort to discarding any. This is now included in the Methods section 2.4.
>
> Finally, we do not expect to find only steady states in nature as multi-stability and environmental fluctuations may vary the observed states of communities [2]; this is a limitation of our current proposal. A potential improvement would be predicting each species' probability distribution (e.g., mean and variance of a Gaussian distribution) and/or predicting several values. Unfortunately, we could not implement these with our datasets due to the lack of samples to estimate variance and due to very few cases of multi-stability.
>
> **W4**
> We updated the Methods accordingly.
>
> **W5-6**
> In our simulations, the interaction matrix $M$ determines the evolution of the community. In principle, this matrix is an unknown function of the genome of different bacterial species. Therefore, while we can define an interesting distribution over interaction matrices, it is extremely challenging to define a distribution over genomes such that this gives rise to a reasonable distribution of interaction matrices. For this reason, we are forced to take an inverse approach to control $M$: we sample interaction matrices, assume a particular form for the function mapping genomes to interactions, and optimize the genome of simulated bacteria to produce the desired interaction matrix. This allows us to indirectly sample from a distribution of genomes such that a function from genomes to the interaction matrix is guaranteed to exist, and the resulting interaction matrix is interesting.
>
> We carefully reviewed our manuscript and did not come across any mention of interpretability for the simulated genomes. Our suggestion for interpreting GNNs and genomes in future work pertains to real data. If there is any confusion, please point it out so we can make appropriate edits.
>
> **W7**
> Thank you for pointing this out; we corrected **invariance** for **equivariance** in the manuscript.
>
> **W8, Q1**
> See the common response; we clarified the Methods. We only use node attributes (the genomes) and, given the lack of prior knowledge on bacterial interactions, we do not use edge attributes and we use a fully connected graph.
>
> **Q2**
> See the common response; we added experiments to quantify the scalability of our method, shown in Appendix B.1 and and S10.
> Our new results demonstrate that our approach can scale to large communities of up to 200 bacteria (Supplementary Fig. S9).
> We also show that given a large number of samples, GNNs’ performance on unseen bacteria increases with the diversity of bacteria seen during training Supplementary Fig. S10). Hence, we believe our method will become increasingly useful with larger-scale datasets.
>
>
> ----
>
> **References**
>
> [1] Lozupone, Catherine A., et al. "Diversity, stability and resilience of the human gut microbiota." Nature 489.7415 (2012): 220-230.
>
> [2] Gonze, Didier, et al. "Multi-stability and the origin of microbial community types." The ISME journal 11.10 (2017): 2159-2166.

---

### Official Review · Reviewer_kpxx · 2023-10-31

**Soundness:** 2 fair
**Presentation:** 2 fair
**Contribution:** 2 fair
**Rating:** 3
**Confidence:** 4

**Summary:**

The paper aims at predicting steady-state composition of microbial communities from the gene content of their genomes using graph neural networks.

**Strengths:**

Understanding how distinct bacteria form communities is an important problem, and the manuscript provides a solid introduction to the topic and, in the experimental section, asks important questions about our ability to understand community formation.

**Weaknesses:**

The proposed approach for using GNNs for bacterial communities is vaguely described and not well justified. The key methods section (2.2) provides a generic description of existing GNN approaches, and is missing key microbiome specific information (in particular, what is the topology of the graph). That information is provided in Supplementary information: the graph is fully connected. This makes statements in the manuscript such as “By using k graph convolutional layers after one another we can achieve k-hop information propagation” rather misleading.

Overall, the proposed method - applying GNN model in a straightforward way to a very small, fully-connected graph - is poorly justified and weak on novelty.

**Questions:**

What is the rationale for using a GNN on a very simple graph?

What is the benefit of focusing on predicting steady state, instead of focusing on dynamical changes to the relative abundances (e.g., dysbiosis).

---

> ### Author Response · Authors · 2023-11-15
> **Response to Reviewer kpxx (part 1/2): motivation to use GNN**
>
> Thanks a lot for your time and feedback. We understand our motivation to use GNNs needs to be clarified and we would like to provide some first answers.
>
>
> - We employ GNNs as a predictive model for microbial systems. Note that this differs from some benchmarks in the GNN literature [1-2], where the goal is to perform e.g. node classification with large graphs with known connectivity as input data.
>
> - We are interested in using the GNN as a dynamics model, where we cannot assume any prior knowledge on the graph connectivity.  Our view of GNNs as dynamics models is similar to the works [3-5]. In all these cases, the graphs are small (3-10 nodes), but still, they benefit from the injection of relational and entity-centric inductive biases via the graph structure, as it offers isolation of information that monolithic networks don’t have [6].
>
> - GNNs offer the advantage of sharing parameters across nodes - treating input genomes as a bag of units rather than 1 concatenated vector, which is key to generalizing to any unseen bacteria and communities of any size. Any new genome can thus be treated by the GNN for predictions without needing to resize the model. Whether the bacterial community is small or big is an external characteristic which does not influence the choice of architecture for this task.
>
> - As rightly pointed out, we do not know the graph topology. It is where GNNs become handy as we can use fully connected graphs that do not make any assumptions about the network. The message passing and/or convolutional layers allow updating nodes based on all members of the community, which is reasonable as all members may affect each other (directly or indirectly [7-8]). The k-hop information propagation should capture k-order relations between entities: the first message passing is limited to the neighboring node attributes (pairwise interactions) and the next ones propagate the interactions of neighbors (bacterium $n_i$ receives information from $n_j$ and how it has been affected by others).
>
>
> We hope these points answer your concerns and that our motivation to use GNN is now more comprehensible; we will edit our manuscript accordingly. Besides, we will answer your 2nd question in the coming days.
>
>
>
> **References**
>
> [1] Sen, Prithviraj, et al. "Collective classification in network data." AI magazine 29.3 (2008): 93-93.
>
> [2] Pei, Hongbin, et al. "Geom-gcn: Geometric graph convolutional networks." arXiv preprint arXiv:2002.05287 (2020).
>
> [3] Sanchez-Gonzalez, Alvaro, et al. "Graph networks as learnable physics engines for inference and control." International Conference on Machine Learning. PMLR, 2018.
>
> [4] Kipf, Thomas, Elise Van der Pol, and Max Welling. "Contrastive learning of structured world models." arXiv preprint arXiv:1911.12247 (2019).
>
> [5] Watters, Nicholas, et al. "Visual interaction networks: Learning a physics simulator from video." Advances in neural information processing systems 30 (2017).
>
> [6] Battaglia, Peter W., et al. "Relational inductive biases, deep learning, and graph networks." arXiv preprint arXiv:1806.01261 (2018).
>
> [7] Pacheco, Alan R., Melisa L. Osborne, and Daniel Segrè. "Non-additive microbial community responses to environmental complexity." Nature communications 12.1 (2021): 2365.
>
> [8] Morin, Manon A., et al. "Higher-order interactions shape microbial interactions as microbial community complexity increases." Scientific Reports 12.1 (2022): 22640.

---

> ### Author Response · Authors · 2023-11-22
> **Response to Reviewer kpxx (part 2/2)**
>
> We thank you again for your feedback. Below is the rest of our response to your concerns and questions. In addition, please check our general response.
>
> **Q1**
> Please, see the 1st part of the response.
> We modified a) our motivation to use GNNs in the Introduction, b) the description of the network topology in the Methods.
>
> **Q2**
> We agree that predicting growth dynamics may be ideal for apprehending bacterial community assembly and interactions. However, in practice, datasets usually contain 1 time point per sample due to time and resource costs. Therefore, approaching the problem via steady-states allows a potential application to many more datasets and applications.
>
> Regarding modeling the dynamics, we argue that directly predicting the quantity of interest in an end-to-end fashion has proven generally more efficient than using intermediates in machine learning. Hence, while our framework could be employed for modeling dynamics recursively, e.g., as in [1-2], it may not necessarily be better.
>
> Finally, understanding dysbiosis events is essential for human health. As of note, dysbiosis is a disruption of the microbiota from a healthy state [3]. Accordingly, dysbiosis can be studied from steady-state data by comparing healthy and diseased samples and does not require a dynamic model.
>
> ----
>
> **References**
>
> [1] Michel-Mata, Sebastian, et al. "Predicting microbiome compositions from species assemblages through deep learning. iMeta 1: e3." (2022).
>
> [2] Baranwal, Mayank, et al. "Recurrent neural networks enable design of multifunctional synthetic human gut microbiome dynamics." Elife 11 (2022): e73870.
>
> [3] Kriss, Michael, et al. "Low diversity gut microbiota dysbiosis: drivers, functional implications and recovery." Current opinion in microbiology 44 (2018): 34-40.

---

### Author Response · Authors · 2023-11-22
**Common response**

We thank the reviewers and program chairs for their time and feedback.

We appreciate that reviewers:
- underlined the importance of modeling growth dynamics of microbial communities (reviewer **kpxx** and **aYM1**) and its challenges (reviewer **aYM1**);
- stressed our efforts to simulate data (reviewers **5GQX** and **U7B8**);
- agreed that GNNs are more suitable for this task than MLPs (reviewer **5GQX**, **aYM1**, and **U7B8**),
- and appreciated our experiments to support our framework (reviewers **aYM1** and **U7B8**).

Using the helpful comments from the reviewers, we edited the manuscript to improve the clarity and presentation (see changes in *green*).
Some reviewer comments led to additional experiments, which we included in the manuscript in *red* and report in the answers below.

We start by answering common questions. Detailed answers are provided in the individual responses.

- There was general confusion about the data (reviewers **5GQX**, **aYM1**): datasets, i.e., Friedman2017, BaranwalClark2022, and each simulation set, were processed independently, meaning that a model was fitted on each but not on all at once. Hence, genome representations are distinct between real data (obtained from genome annotations) and simulated ones (resulting from our simulator). Genomes, i.e., node features, are always binary vectors. We edited the Method section to clarify this point.
- One primary concern was the scalability of the method (reviewers **5GQX**, **aYM1**). We have now conducted experiments that we report in Appendix B and figures S9 and S10 to analyze the effect of:
    1. larger communities of up to 200 bacteria;
    2. the training set size, i.e., the number of communities seen during training;
    3. the diversity in the training set as a mean to improve generalization to unseen bacteria.

In addition, we enhanced our results by:
- Increasing the number of model seeds from 3 to 5;
- Plotting the average $R^2$ with a 95% confidence interval across all samples calculated on sample bootstraps using the model ensemble (instead of plotting the $R^2$ per model seed and for the model ensemble);
- Changing the MPGNN node update function to match the C-SWM in [1].
We also show these changes in *red*.
If there are remaining issues or points to clarify, we would be happy to address them.

----

**References**

[1] Kipf, Thomas, Elise Van der Pol, and Max Welling. "Contrastive learning of structured world models." ICLR (2020).

---

### Meta-Review · Area_Chair_hyPi · 2023-12-05

**Metareview:**

The paper studies bacterial communities by applying GNNs. All reviewers generally agreed that the area is interesting, but raised a bunch of questions about the methodology, the details, scalability, and more. Just as one example, it's not clear why a fully-connected graph makes sense here (in the absence of more specific knowledge on the community graph). While the authors provided some answers, ultimately there's enough outstanding questions and unclear items to make me believe the paper is still below the bar in the current iteration. However, I am optimistic about a future draft.

**Justification For Why Not Higher Score:**

Lack of methodological clarity.

**Justification For Why Not Lower Score:**

N/A

---

### Decision · Program_Chairs · 2024-01-16

Reject